# Direct conversion of cardiac fibroblasts into endothelial-like cells using *Sox17* and *Erg*

Gregory Farber [1], Yanhan Dong[1], Qiaozi Wang [1], Mitesh Rathod[2,3], Haofei Wang [1], Michelle Dixit [1], Benjamin Keepers [1,4], Yifang Xie [1], Kendall Butz[1], William J. Polacheck [1,2], Jiandong Liu [1,4] & Li Qian [1,4] ✉

Endothelial cells are a heterogeneous population with various organ-specific and conserved functions that are critical to organ development, function, and regeneration. Here we report a *Sox17-Erg* direct reprogramming approach that uses cardiac fibroblasts to create differentiated endothelial cells that demonstrate endothelial-like molecular and physiological functions in vitro and in vivo. Injection of these induced endothelial cells into myocardial infarct sites after injury results in improved vascular perfusion of the scar region. Furthermore, we use genomic analyses to illustrate that *Sox17-Erg* reprogramming instructs cardiac fibroblasts toward an arterial-like identity. This results in a more efficient direct conversion of fibroblasts into endothelial-like cells when compared to traditional *Etv2*-based reprogramming. Overall, this *Sox17-Erg* direct reprogramming strategy offers a robust tool to generate endothelial cells both in vitro and in vivo, and has the potential to be used in repairing injured tissue.

Cardiac disease remains a global malady that requires new innovations to prevent and treat tissue damage. Regenerating new tissue not only requires the production of necessary specialized cardiac cells but also the functional integration of these cells with the surrounding tissue. The generation of new cardiac endothelial cells is particularly challenging, as these new cells must not only provide proper cues for new tissue formation in the cardiac-specific microenvironment but also engraft with pre-existing vasculature. Furthermore, organ-specific endothelial heterogeneity makes it crucial to create a properly specialized endothelial cell equipped to support healthy organ function.

Direct reprogramming is a promising approach to generate the cells necessary to repair injured cardiac tissue. Conversion of cardiac fibroblasts into induced-cardiomyocytes via direct reprogramming has yielded both cardiomyocyte-like cells and reduced fibrosis, resulting in improved cardiac function after in vivo myocardial infarction[1,2]. Reprogramming has also proven to be a viable methodology to generate endothelial cells from non-endothelial cell types[3–8]. Current transcription factor-based reprogramming approaches have

centered around using *Etv2*, a transcription factor involved in the differentiation of immature endothelial cells from angioblasts[9]. Prior studies have illustrated that *Etv2* alone, as well as with additional transcription factors, is able to generate induced endothelial cells (iECs). However, these traditional methodologies report a wide range of reprogramming efficiencies depending on the starting cell type, and they often require additional purification steps or reinfection of the reprogramming cocktail to aid in the generation of iECs and maintain the induced endothelial identity[4,8]. Since endothelial cells further differentiate after the *Etv2*-mediated cell fate decision during embryonic development, and *Etv2* is not expressed in adult endothelial cells, we sought a direct reprogramming approach that involves factors found in adult cardiac endothelium, targets a fully differentiated endothelial cell fate, and is applicable to organ-specific fibroblasts.

In this study, we screened both previously published and other genes that are associated with developing and mature cardiac endothelium and found that the retroviral-induced expression of *Sox17* and *Erg* directly reprograms murine cardiac fibroblasts into induced-

[1]The McAllister Heart Institute, The University of North Carolina at Chapel Hill, Chapel Hill, NC 27599, USA. [2]Joint Department of Biomedical Engineering, University of North Carolina at Chapel Hill and North Carolina State University, Chapel Hill and Raleigh, NC, USA. [3]University of North Carolina Kidney Center, Chapel Hill, NC, USA. [4]Department of Pathology and Laboratory Medicine, The University of North Carolina at Chapel Hill, Chapel Hill, NC 27599, USA. ✉e-mail: li_qian@med.unc.edu

endothelial cells. The created iECs display the functional properties of endothelial cells, interact with cardiomyocytes, and improve vascular perfusion of myocardial scar area in adult mouse hearts post-myocardial infarction when injected at the time of injury. Genomic analyses revealed that this reprogramming strategy generates endothelial cells that are more arterial in nature based on marker gene expression and the mechanism in which they transition from fibroblasts to endothelial cells. When compared to *Etv2*-based reprogramming, this reprogramming mechanism instructs the targeted fibroblasts to a differentiated endothelial identity, whereas traditional *Etv2* reprogramming directs the targeted fibroblasts to an immature endothelial state. Furthermore, this more direct approach allows for more efficient generation of iECs in both neonatal and adult cardiac fibroblasts as well as lung and skeletal muscle fibroblasts when compared to *Etv2*-based reprogramming. This study also demonstrates the in vivo ability of this *Sox17-Erg* retroviral-based approach to convert native murine cardiac fibroblasts present in infarcted myocardium into endothelial cells.

## Results

### Sox17 and Erg reprogram murine cardiac fibroblasts

We designed a screening strategy to identify the reprogramming factors that best convert neonatal cardiac fibroblasts into endothelial cells by combining the methodologies used in MGT-mediated iCM reprogramming and published strategies for iEC reprograming (Fig. 1a). Murine cardiac neonatal fibroblasts were first isolated and then infected with retrovirus on the following day. The cells were then stained and quantified for PECAM1 expression 7 days after retroviral infection. We analyzed previously published scRNAseq datasets from E12.5 developing coronary endothelial cells and adult murine heart cells to contextualize the previously published reprogramming factors and to identify new potential factors for reprogramming (Fig. 1b)[10,11]. All of the identified factors except *Etv2* were found to be expressed in at least one cluster of the developing heart endothelial cells, an expected finding given that *Etv2* targets a cell fate decision prior to the establishment of specialized endothelial identity[9] (Fig. 1c, e). Analysis of the expression of these factors in the adult heart also illustrated the lack of *Etv2* expression in adult heart cells, which is predominantly restricted to the testes of adult mice[9]. Of the selected genes, only *Sox17* and *Erg* were found to be specifically expressed in adult cardiac endothelial and endocardial cells. In contrast, other genes were highly expressed in endothelial cells but also expressed in other cell types, such as *Nr2f2* and *Fli1* (Fig. 1d, f). Ten factors were selected and screened by pooling all 10 genes and then removing one factor to determine the most essential component. Removal of *Sox17* had the most drastic negative effect on the number of PECAM1-positive cells (Fig. 1g, average of 3.8% for 10F to 1.4% with removal of *Sox17*). With *Sox17* identified as the most crucial factor, cells were then reprogrammed with either *Sox17* alone or with an additional factor to determine whether an additional factor increases the number of generated PECAM1-positive cells. Four potential candidates (*Erg, Etv2, Fli1*, and *Klf2*) were identified as having a positive impact on *Sox17*-mediated iEC generation (Fig. 1h, average of 2.6% for *Sox17* alone to >7% for potential candidates). These five factors were then pooled and single constructs were removed to determine whether removal of any of the non-*Sox17* constructs would impact reprogramming. Notably, only the removal of *Etv2* resulted in the increase of PECAM1-positive cells relative to the five-factor combination (Fig. 1i, average of 7.3% for 5F to 10.8% with removal of *Etv2*). Finally, the remaining possible combinations of reprogramming cocktails were compared to *Sox17* alone, as well as the previously published cocktail *Etv2 Erg Fli1*. The addition of *Erg* to *Sox17* resulted in the simplest reprogramming cocktail comprised of the fewest factors that resulted in the greatest number of PECAM1-positive cells (Fig. 1j–n, Supplementary Fig. 1, 5.6% for *Etv2 Erg Fli1* and 22.5% for *Sox17 Erg*).

### Sox17-Erg reprogramming generates functional iECs

With *Sox17-Erg* creating PECAM1-expressing cells by Day 7 of reprogramming, we further characterized the endothelial molecular function and marker expression of these cells and investigated when these cells acquire and maintain endothelial cell identity. A PECAM1-positive cell depletion step was added to the fibroblast isolation protocol in order to limit the potential impact of primary neonatal endothelial cell contamination on the characterization of iECs (Fig. 2a, Supplementary Fig. 2). Flow cytometry analysis of Day 7 samples demonstrated that *Sox17-Erg* reprogramming resulted in a greater number of PECAM1-positive CDH5-positive cells (average of 12.57%) than both *Etv2* (4.03%) and negative control (0.76%) samples (Fig. 2b, e). *Sox17-Erg* samples continued to have a greater number of PECAM1 CDH5 dual positive cells at both 2 weeks and 4 weeks than control samples (Fig. 2c–e). Immunofluorescence analysis of day 7 and 4-week samples further validated our flow cytometry findings and demonstrated that iECs expressed two pan-endothelial marker genes, CDH5 and PECAM1, at day 7 through 4 weeks post-retroviral infection at significantly higher percentages when compared to control samples. Of note, iECs formed cord-like structures by Day 7 when cultured on collagen-coated dishes and when plated on Matrigel for a cord-formation assay similar to what is observed in cultured primary endothelial cells[12] (Fig. 2f, g, Supplementary Fig. 3d). Beyond the cord formation, the *Sox17-Erg* generated iECs demonstrated additional endothelial cell functional properties in vitro. iECs uptake Ac-LDL and produce nitric oxide at higher levels than control fibroblast cells (Fig. 2h, i, Supplementary Fig. 3a, b). iECs also responded to TNFalpha stimulation in a similar fashion to native endothelial cells, as seen by increased expression of *Icam1, Sele*, and *Vcam1* relative to their unstimulated condition (Fig. 2j). Interestingly, this ability of iECs to respond to TNFalpha may be inherited from their initial fibroblast identity since fibroblasts can also respond to TNFalpha. However, stimulated-iECs responded slightly differently in *Icam1* and *Vcam1* expression when compared to control cells. The resulting TNFalpha stimulation of iECs also corresponded to binding THP1 monocytes at higher levels than without TNFalpha stimulation, further confirming their endothelial-like functionality in inflammatory conditions (Fig. 2k, Supplementary Fig. 3c).

Bulk RNAseq was performed on Day 3, Day 7, and 4-week control and *Sox17-Erg* samples to further determine which genes were activated by *Sox17-Erg* reprogramming and to identify when their endothelial identity was acquired and maintained. Comparison of the different conditions illustrated that the *Sox17-Erg* samples established a distinct identity by Day 3 and maintained an endothelial-like identity into 4 weeks, supporting our functional analyses that utilize 4-week-old iECs (Fig. 2l). Clustering of the differentially expressed genes (DEGs) at each timepoint revealed three distinct gene groups. The control gene cluster DEGs were more lowly expressed by Day 3 in *Sox17-Erg* samples and remained expressed in the control samples. *Sox17-Erg* Cluster#1 pertained to genes that were more highly expressed by Day 7 and maintained their expression by 4 weeks. These genes included canonical endothelial cell marker genes (i.e. *Pecam1, Cdh5, Vwf* and *Tek*), and artery development and endothelial cell migration were among the top GO Biological Process terms when these genes were analyzed by enrichment analysis (Fig. 2m, Supplementary Fig. 4a). The second set of *Sox17-Erg* differentially expressed genes was highly expressed by Day 3 but later more lowly expressed and related to activities in the nucleus suggesting transcriptional changes but endothelial identity is not yet established. The acquisition and maintenance of endothelial identity and function were further defined by GO Molecular Function enrichment of the unique *Sox17-Erg* differentially expressed genes at each timepoint. At Day 3 the enrichment terms were related to activity in the nucleus, which transitioned to receptor activity at Day 7 and then centered around growth factor binding and cell adhesion molecules at 4 weeks

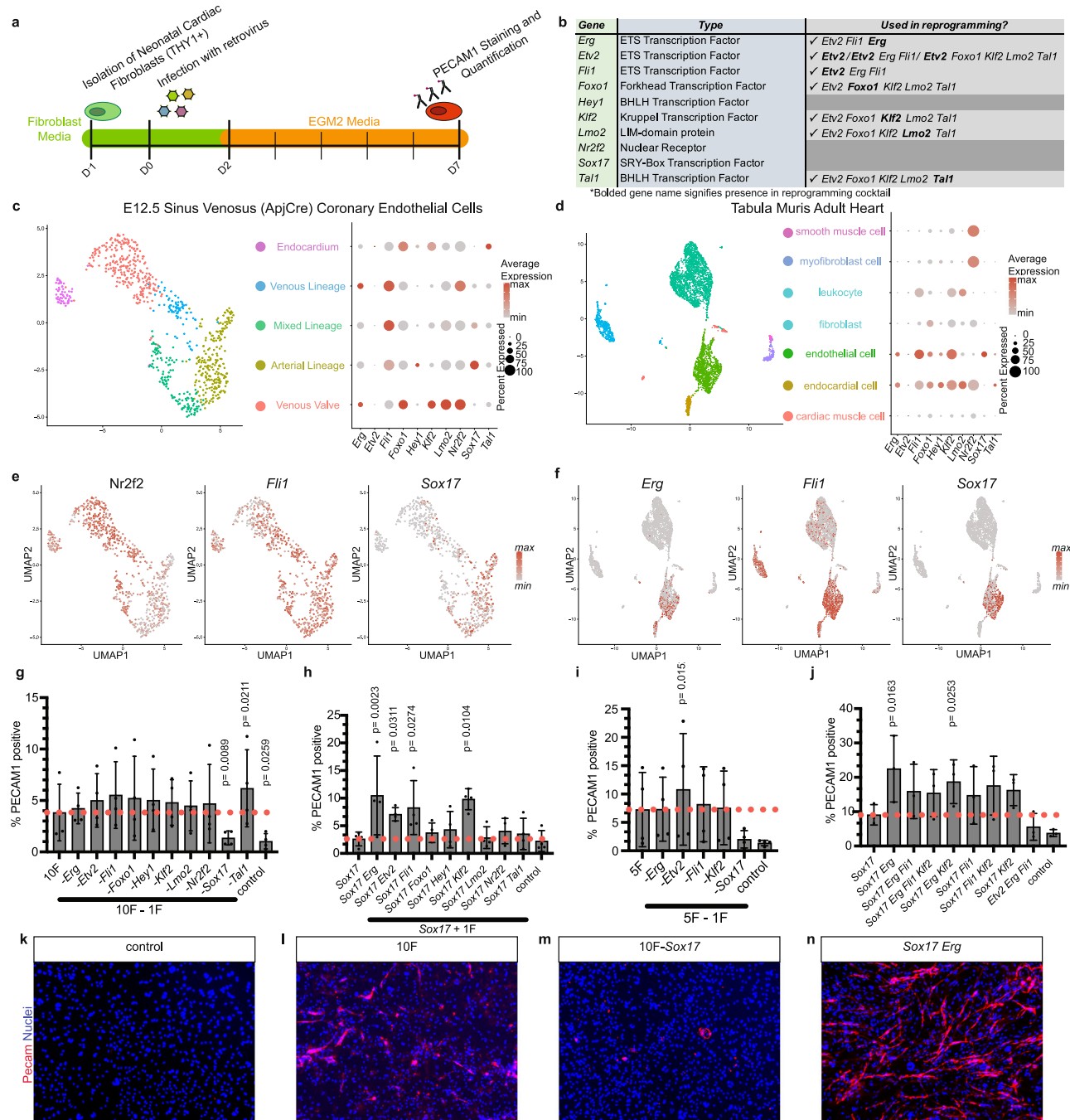

**Fig. 1 | Sox17 and Erg reprogram murine cardiac fibroblasts. a** Schematic of screening factors that impact the direct reprogramming of cardiac fibroblasts to iECs. **b** Table of previously published and other factors to be tested in iEC direct reprogramming. **c** UMAP of E12.5 coronary endothelial cells with dot plot of selected factor gene expression. **d** UMAP of Tabula Muris Adult Heart cells with dot plot of selected factor gene expression. **e** Expression profiles of *Nr2f2*, *Fli1*, and *Sox17* in E12.5 murine cardiac endothelial cells. **f** Expression profiles of *Erg*, *Fli1*, and *Sox17* in adult murine cardiac cells. **g**–**j** Quantification of day 7 PECAM1 immunofluorescence screening data (two-sided ratio paired Student's *t*-test, mean with SD) (**g**–**i** n = 4 independent samples, **j** n = 3 independent samples). **k**–**n** Representative day 7 images of screening selected reprogramming factors (scale bars 275 μm). Source data are provided as a Source Data file.

(Fig. 2n). Cell type enrichment analysis of the differentially expressed gene groups further confirmed the acquisition of endothelial cell identity. The *Sox17+Erg* Gene Cluster#1 genes were similar to other published endothelial cell gene sets whereas the control cluster genes enriched for more fibroblast-like cell types (Fig. 2o). Thus, by Day 7 of *Sox17-Erg* reprogramming, the targeted cells converted to an endothelial identity that is maintained for at least four weeks and iECs displayed canonical endothelial cell functionality and marker expression in vitro.

## iECs engraft into infarcted tissue

To improve reprogramming efficiency and fluorescently label reprogrammed cells, we created two polycistronic constructs that contain a GFP-tag, *Sox17* and *Erg* (Fig. 3a)[13,14]. The positions of *Sox17* and *Erg* were swapped in these constructs to determine whether the relatively higher expression of either factor impacts the number of PECAM1-expressing cells on Day 7 (Supplementary Fig. 4b). *Sox17* in the first position resulted in a greater number of PECAM1-positive cells at Day 7 (Fig. 3c and d, average of 18.5% for ESG and 27.3% for SEG). In addition,

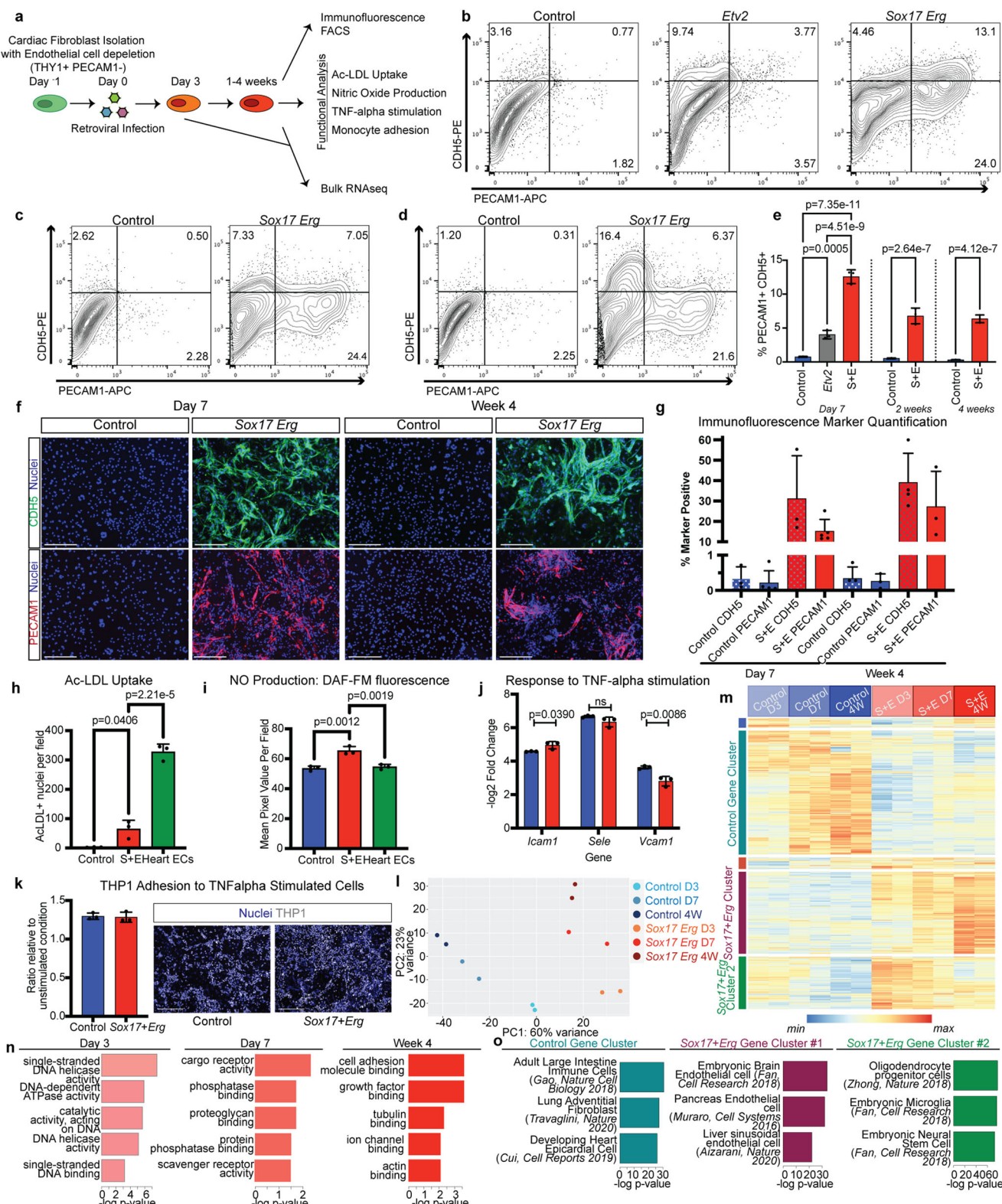

the *Sox17-Erg-GFP* (SEG) construct was also more efficient in converting infected cells, with an average conversion rate of 52.3% compared to only 35.3% with *Erg-Sox17-GFP* (Fig. 3b, d).

Now that we were able to trace which fibroblasts are infected with the reprogramming construct, we then mixed Day 7 iECs, non-reprogrammed cardiac fibroblasts, or native endothelial cells with isolated primary murine cardiomyocytes at a ratio of one to one to evaluate whether direct reprogramming resulted in endothelial-

cardiomyocyte interactions that occur in vivo. Remarkably, when cultured with iECs and observed 5 days after cell plating, the cardiomyocytes were found to be forming GJA1 junctions with the iECs at a similar percentage to both native fibroblasts and cardiac endothelial cells (Fig. 3e, f).

Given that the iECs readily formed tubes and displayed other endothelial functional properties, we tested whether this reprogramming conferred the ability to withstand shear stress due to laminar

**Fig. 2 | Sox17-Erg reprogramming generates functional iECs. a** Workflow schematic of the generation of and functional characterization of iECs. Representative CDH5 PECAM1 flow cytometry analysis of Day 7 (**b**), 2 weeks (**c**), and 4 weeks (**d**) samples. **e** Graph of CDH5 PECAM1 double positive cells from flow cytometry analysis (one-way ANOVA with Tukey post-test, *n* = 3 independent samples, mean with SD). **f** Immunofluorescence staining of 7-day and 4-week control and iEC samples (scale bars 275 μm). **g** Quantification of CDH5 and PECAM1 immunofluorescence data (mean with SD) (PECAM1 day 7 *n* = 5 independent samples, PECAM1 week 4 *n* = 3 independent samples, CDH5 day 7 *n* = 3 independent samples, CDH5 week 4 *n* = 4 independent samples). **h** Quantification of Ac-LDL Uptake of 4-week-old samples (*n* = 3 independent samples) (one-way ANOVA with Tukey post-test, mean with SD). **i** Quantification of DAF-FM signal intensity to detect NO production (*n* = 3 independent samples) (one-way ANOVA, mean with SD). **j** qPCR of TNFalpha stimulation response genes relative to unstimulated condition (control in blue, *Sox17-Erg* in red) (*n* = 3 independent samples) (two-sided Student's *t*-test, mean with SD). **k** Quantification and representative images of THP1 cell adhesion to stimulated control and *Sox17-Erg* iECs (*n* = 3 independent samples, mean with SD) (scale bars 275 μm). **l** Principal component analysis of bulk RNAseq control and *Sox17-Erg* iEC samples. **m** Heatmap of differentially expressed genes identified between control and *Sox17-Erg* iEC samples with further characterization of the identified gene clusters using GO Biological Process enrichment. **n** GO Molecular Function enrichment of differentially expressed genes specific to the age of iEC samples (one-sided Fisher's exact test with Benjamini–Hochberg correction). **o** Enrichment comparison analysis of identified differentially expressed gene clusters found in **k** to previously published datasets (one-sided Fisher's exact test with Benjamini–Hochberg correction). Source data are provided as a Source Data file.

flow and whether the iECs aligned with the direction of flow like native endothelial cells. Since our bulk RNAseq data suggested that these iECs are potentially more arterial in nature, we subjected iECs to higher shear stresses that are expected in arterial and capillary vessels. After 24 h of flow at 35 dynes/cm$^2$ of shear stress, iECs began to align with and elongate parallel to the direction of flow (Fig. 3g–i). Furthermore, we observed a greater percentage of GFP-positive cells in the flow condition, suggesting that non-reprogrammed fibroblasts did not survive the laminar flow and that reprogramming confers the ability for the reprogrammed cells to withstand the shear stresses normally found in capillaries and arteries in vivo (Fig. 3j).

To determine whether the ability to interact with cardiomyocytes and to survive laminar flow in vitro translates to in vivo functionality, we evaluated whether the iECs could engraft with host tissue by injecting one million SEG-generated Day 7 iECs into infarcted regions of SCID murine hearts. All mice that received injections of iECs showed engraftment of iECs when the hearts were harvested for analysis seven days post-injection. Three-dimensional confocal imaging revealed that the injected cells were able to attach to native endothelial cells in myocardial scar regions (Fig. 3k, l, Supplementary Fig. 5).

The presence of PECAM1-positive iECs on Day 7 led us to evaluate the impact of iEC injection on cardiac function and scar vessel perfusion at four weeks post-injury. Injection of iECs did not impact cardiac function measured by ejection fraction or fractional shortening when compared to PBS control samples (Fig. 3m, Supplementary Fig. 5c). We did observe a positive trend in the percent area of lectin perfused vessels found in the iEC infarct injury area (average of 10.83%) when compared to PBS control hearts (average of 4.99%) (Fig. 3n, o). However, when the lectin-perfused volume of a scar region was normalized to a non-scar region of the same sample to take into account the impact of myocardial injury on overall cardiac vessel perfusion, we observed a statistically significant increase in iEC-injected mice (Fig. 3p, q, Supplementary Fig. 5d).

## Sox17 Erg generate a distinct type of iECs

With *Sox17 Erg* being able to generate functional endothelial cells without *Etv2*, we performed scRNAseq of *Sox17 Erg* reprogramming in parallel with the traditional *Etv2* reprogramming to compare the mechanisms of these two reprogramming cocktails. Single-cell RNA sequencing was performed on *Etv2*, *Sox17 Erg*, and control samples that were pooled and multiplexed at both Day 3 and Day 7 timepoints (Fig. 4a). After the samples were demultiplexed, the captured cells were filtered based on quality control metrics, merged, and clustered resulting in 10,023 cells (Supplementary Fig. 6). Uniform manifold approximation and projection (UMAP) of the samples revealed that all three sample types clustered distinctly (Fig. 4b, c). Four clusters could be directly linked to *Etv2* reprogramming, while *Sox17-Erg* had five associated clusters. Plotting of cell expression of fibroblast (*Thy1* and *Tcf21*) and endothelial genes (*Cdh5*, *Vegfr2*, *Pecam1*) illustrated the relative downregulation of fibroblast genes and upregulation of

endothelial genes in both reprogramming methodologies when compared to the control condition (Fig. 4d–h). All clusters demonstrated unique marker gene expression signatures, and several of the *Etv2* and *Sox17 Erg* top cluster genes were endothelial-associated, such as *Emcn* for *Etv2*#3 and *Cav1* for *Sox17 Erg*#2 (Fig. 4i, Supplementary Data 1). To further compare the differently generated iEC cluster types, GO-term analysis of the marker genes revealed that two clusters for both *Etv2* and *Sox17 Erg* best represented acquired endothelial identity based on enrichment for endothelial cell development and endothelial cell migration terms (Fig. 4j). The marker genes for these clusters were then analyzed using KEGG Enrichment, and *Etv2*#2 and *Sox17-Erg*#2 were selected for further investigation based on the enrichment for focal adhesion and fluid shear stress and atherosclerosis (Fig. 4k). When compared to published gene sets, both iEC gene sets enriched for endothelial subtypes. However, *Etv2*#2 was more similar to hepatic sinusoidal endothelial cells whereas the marker genes for *Sox17 Erg*#2 were more similar to hepatic microvascular endothelial cells and brain endothelial cells (Fig. 4l). These results suggested that the iECs generated by both strategies are distinct and provided preliminary evidence that the two mechanisms for the generation of these iECs are different. Because our aim was the generation of mature cardiac endothelial cells, we annotated the identities of individual cells based on the Tabula Muris Adult Heart scRNAseq dataset used in the initial screening of the *Sox17-Erg* cocktail[11]. Both *Etv2* and *Sox17-Erg* generated cells that were annotated to cardiac endothelial cells with clusters *Etv2*#2, *Sox17 Erg*#2, and *Sox17 Erg*#3 being the primary clusters with cells identified as cardiac endothelial cells (Fig. 4m). Interestingly, we observed a dramatic increase in the density of G2M stage cells in the *Sox17-Erg* Day 3 cells when compared to control and *Etv2* Day 3 cells (Fig. 4n). This initial increase of *Sox17-Erg* cells in the G2M stage followed by transitioning to G1/G0 is comparable to what has been reported in the acquisition of arterial identity in the developing mouse retinal and cardiac vasculature[10,15,16]. Furthermore, this cell stage transition paired with unique cluster identities highlights the acquisition of a differentiated, most likely arterial, endothelial identity in *Sox17 Erg* reprogrammed cells. The differences in marker expression, differentiated identity, and cell cycle states led us to investigate whether there are differences in cell signaling in the context of this in vitro data (Fig. 4o). Interestingly, all *Etv2*-related clusters appeared primed to receive signals rather than send them while *Sox17 Erg* clusters were predicted to send more signals rather than receive (Supplementary Fig. 7).

The difference in the types of generated iECs led us to explore whether the starting identity of the reprogrammed cells affects the reprogramming of either approach. To test this, we reprogrammed fibroblasts isolated from 3-month-old murine hearts with either *Etv2* or SEG. Adult cardiac fibroblasts were chosen because they are more differentiated than neonatal fibroblasts and therefore less plastic, and the age of the cells has been shown to affect traditional *Etv2*-based reprogramming[8,17]. SEG reprogramming was able to generate 4-fold greater number of PECAM1-positive iECs by Day 7 when compared to

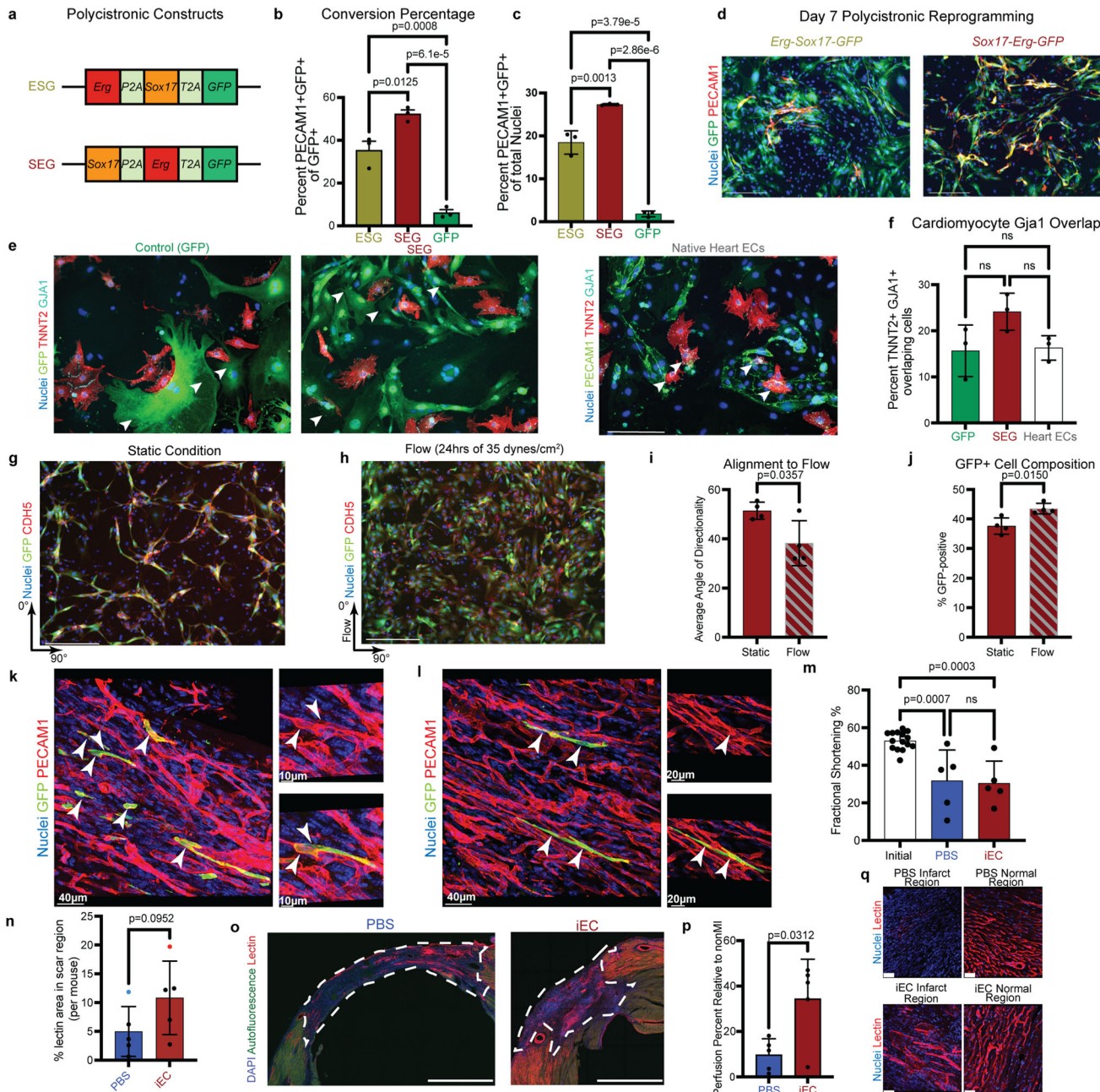

**Fig. 3 | iECs engraft into infarcted tissue. a** Polycistronic constructs containing *Sox17*, *Erg*, and *GFP*. **b** Quantification of the percent of cells that are infected with polycistronic construct and are PECAM1-positive relative to the number of infected cells at day 7 (*n* = 3 independent samples) (one-way ANOVA with Tukey post-test, mean with SD). **c** Quantification of PECAM1-positive cells at day 7 of reprogramming (*n* = 3 independent samples) (one-way ANOVA with Tukey post-test, mean with SD). **d** Representative images of day seven reprogramming of polycistronic constructs (scale bar 275 µm). **e** Representative images of cardiac fibroblast, iEC, or neonatal cardiac endothelial cells cultured with neonatal cardiomyocytes with white arrows indicating cardiomyocytes interacting with non-cardiomyocytes (scale bar 150 µm). **f** Quantification of percent cardiomyocytes with GJA1+ interactions with non-cardiomyocytes (*n* = 3 independent samples) (one-way ANOVA with Tukey post-test, mean with SD). **g** Day 7 iECs cultured under static conditions (scale bar 275 µm). **h** iECs cultured for 24 h at 35 dynes/cm² flow (scale bar 275 µm). **i** Quantification of GFP alignment to the direction of flow (*n* = 4 independent samples) (two-sided Student's *t*-test, mean with SD). **j** Quantification of percent

GFP-positive cells after static or flow conditions (*n* = 4 independent samples)(two-sided Student's *t*-test, mean with SD). **k, l** Deconvolved three-dimensional images of infarcted murine heart regions that contain SEG iECs that engrafted into pre-existing vasculature (arrows indicate GFP+ PECAM1+ SEG iECs). **m** Fractional shortening of heart before and 4 weeks post myocardial infarction (initial *n* = 15 animals, iEC and PBS *n* = 5 animals) (one-way ANOVA with Tukey post-test, mean with SD). **n** Comparison of percent lectin perfusion post myocardial infarction of scar region (*n* = 5 animals) (two-sided Mann–Whitney test; mean with SD; selected samples for representative images are blue (PBS) or red (iEC)). **o** Representative maximum intensity projection images of lectin-perfused PBS and iEC scar regions (scar region outlined with dashed line). **p** Comparison of normalized percent lectin perfusion in scar area to non-scar area (*n* = 5 animals)(two-sided Student's *t*-test, mean with SD). **q** Representative three-dimensional images of lectin perfused injury and non-injury areas (scale bar 40 µm). Source data are provided as a Source Data file.

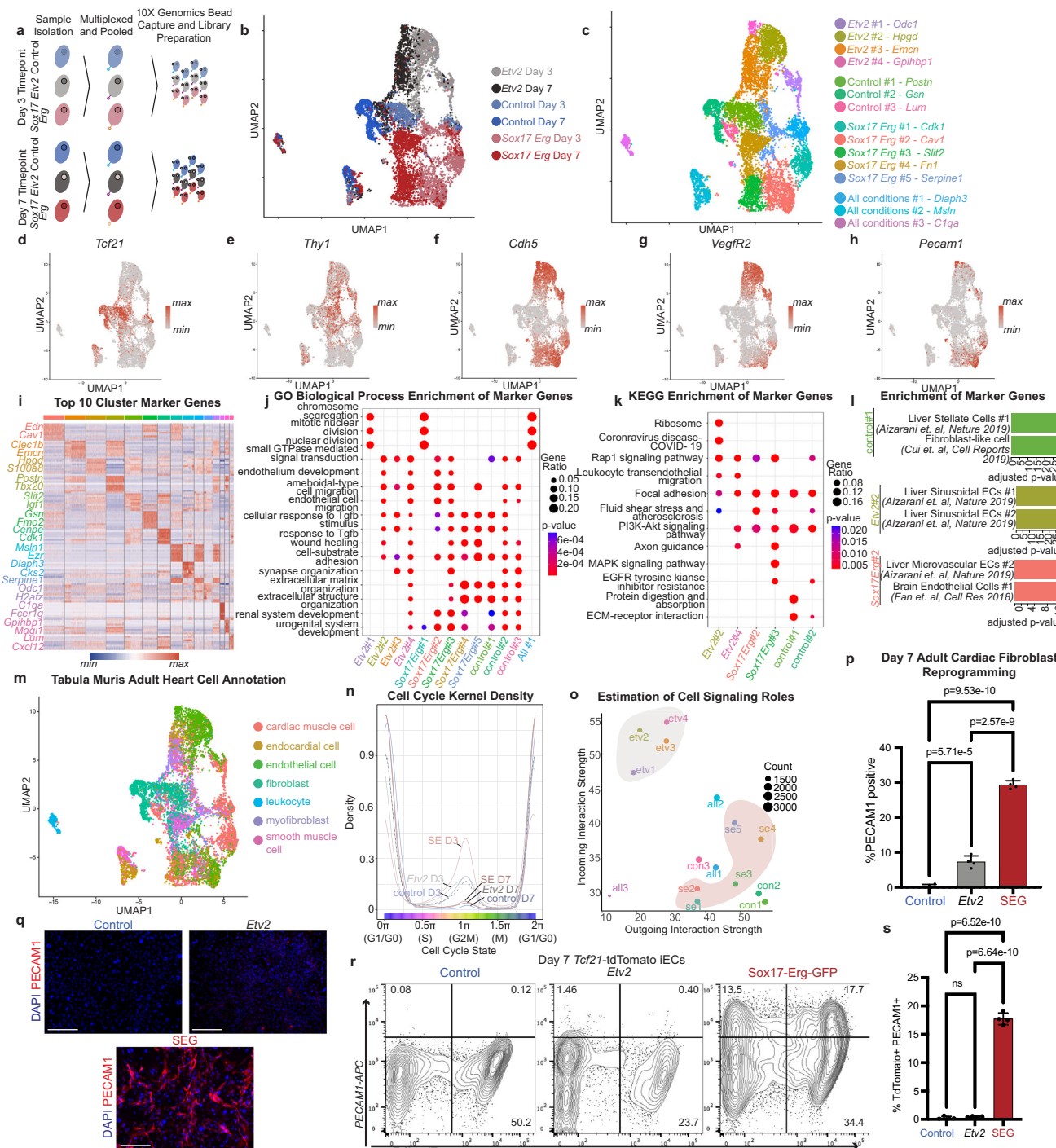

**Fig. 4 | Sox17 and Erg generate a distinct type of iEC. a** Schematic of multiplexing and pooling scRNAseq samples. **b** UMAP of scRNAseq data annotated by sample type. **c** UMAP of scRNAseq data annotated by cluster names. **d, e** Expression profiles of fibroblast marker genes (*Tcf21* and *Thy1*). **f–h** Expression profiles of endothelial marker genes (*Cdh5*, *Vegfr2*, *Pecam1*). **i** Heatmap of top 10 cluster-specific marker genes. **j** Plot of GO Biological Process enrichment of cluster-specific marker genes (one-sided Fisher's exact test with Benjamini–Hochberg correction). **k** Plot of KEGG enrichment of cluster-specific marker genes of a subset of clusters identified in **j**. **l** Enrichment analysis of top endothelial-like clusters (*Etv2*#2 and *Sox17 Erg*#2) and control sample to previously published gene sets (one-sided Fisher's exact test with Benjamini–Hochberg correction). **m** Gene set annotation of iEC and control

samples to Tabula Muris Adult Heart. **n** Cell cycle density graph, annotated by sample type, illustrating the proportion of cells predicted to be in each stage of the cell cycle. **o** Incoming and outgoing estimated cell signaling roles of identified cell clusters. **p** Quantification of ratio of PECAM1-positive cells of day 7 adult cardiac fibroblast reprogramming (*n* = 4 independent samples) (one-way ANOVA with Tukey post-test, mean with SD). **q** Representative images of day 7 reprogramming of adult murine cardiac fibroblasts (scale bar 150 μm). **r** *Tcf21*-tdTomato PECAM1 Day 7 flow cytometry analysis of lineage-traced adult cardiac fibroblasts. **s** Comparison of Day 7 *Tcf21*-tdTomato PECAM1 double positive cells from control, *Etv2*, and SEG conditions (*n* = 4 independent samples) (one-way ANOVA with Tukey post-test, mean with SD). Source data are provided as a Source Data file.

*Etv2* (Fig. 4p, q). To further validate that the generation of the iECs is due to the conversion of fibroblasts into endothelial cells, we utilized a *Tcf21* tamoxifen-inducible CRE-recombinase to label the *Tcf21* fibroblast lineage with tdTomato in adult mice. The adult cardiac fibroblasts of these lineage-traced mice were then plated and isolated using THY1-positive MACS selection 7 days after the last tamoxifen dose. Flow cytometry analysis of Day 7 reprogrammed and control cells revealed that SEG iEC reprogramming led to a statistically significant increase in PECAM1-positive *Tcf21*-tdTomato-positive cells than both *Etv2* and negative samples. Interestingly, we detected no significant increase in the amount of these double-positive cells in the *Etv2* samples relative to the negative control. This difference in reprogramming efficiency in isolated adult cardiac fibroblasts led us to evaluate whether SEG is more efficient in reprogramming fibroblasts from other adult organs into iECs. Using the THY1-positive MACS isolation methodology, we isolated fibroblasts from adult murine skeletal muscle (quadriceps muscle) and lung. For both skeletal muscle and lung iECs, SEG resulted in a greater number of PECAM1-positive cells at Day 7 than *Etv2* (Supplementary Fig. 8). This ability for *Sox17-Erg* to reprogram adult fibroblast from multiple organs at a higher rate than *Etv2* further illustrates the potency of *Sox17-Erg* direct reprogramming.

### Sox17-Erg reprogramming navigates a different route

The independent clustering of control, *Etv2*, and *Sox17-Erg* samples and the distinct types of endothelial cells generated by the two different strategies led us to perform pseudotime analysis of the scRNAseq data, which revealed three trajectories specific to each sample type (Fig. 5a). We isolated each trajectory and found the gene modules that best described the cell fate transitions of each condition (Fig. 5b, c). When condition-specific gene modules were split into three stages and analyzed by GO Molecular Function enrichment, both *Etv2* and *Sox17-Erg* had similar first and second stages (Fig. 5d). Despite the similar GO-term enrichment, the trajectories diverged by Stage#2 seen by cluster-specific expression of module genes (Fig. 5e, f). By stage two of *Etv2* reprogramming, unique genes were activated that were not highly expressed in *Sox17-Erg* clusters, and the *Etv2* discrete identity was less clear by stage 3 (Fig. 5e). For *Sox17-Erg* reprogramming, the pseudotime progression was even starker with the stage three genes being specific to Sox17-Erg#3 cluster and lowly expressed in the *Etv2* clusters. Comparison of cluster-marker genes and the pseudotime gene modules identified genes that provided further insight into the mechanisms and the targeted cell fate of these two reprogramming strategies (Fig. 5f). While both *Etv2* and *Sox17-Erg* activated a generic endothelial gene program seen by *Cdh5* expression or marker gene expression (*Hpgd* and *Igfbp3*), *Etv2* directed the targeted fibroblasts to an immature endothelial fate seen by the activation and maintained expression levels of *Vegfr3* (Fig. 5g). The expression of *Vegfr3* was of particular note because it is normally expressed in developing coronary vessels and lymphatic vessels and its expression is shut off in properly matured adult coronary vessels[18–20]. *Sox17-Erg*, however, directly targeted a differentiated arterial fate as seen by the gradual decrease in *Cdk1* expression and increase of *Gja5*, both of which mimic what has been observed in the development of pre-artery and arterial vessels[10,21,22] (Fig. 5g). This decrease in *Cdk1* expression matched what we observed in the shift of distribution of cell cycle states at Day 3 versus Day 7 (Fig. 4n). Furthermore, *Sox17-Erg* appeared to skip the intermediate immature state by not expressing *Vegfr3* during its pseudotime trajectory while *Etv2* iECs did not activate the arterial programming genes (Fig. 5g). This potentially signifies that *Sox17-Erg* directly targets a terminally differentiated fate while *Etv2* directs towards an intermediary precursor stage.

### Sox17-Erg activates differentiated endothelial identity

Given the differences in the two trajectories and end products of these reprogramming approaches, we performed H3K27ac CUT&Tag on Day

3 samples to determine the genes activated by these cocktails and to compare them to control fibroblasts and isolated neonatal cardiac endothelial cells[23] (NCEC). The peaks specific to each condition were determined by identifying the differentially expressed peaks relative to those in control fibroblast samples. As expected, sample-specific peak sets showed low enrichment in control samples, and both the *Etv2*-iEC and SEG-iEC samples showed enrichment for the primary endothelial cell peaks (Fig. 5h, Supplementary Fig. 9). Additionally, the *Etv2* and SEG-specific peaks were found to be enriched in the primary endothelial cell samples. Comparison of the annotated peak genes illustrated that many genes were shared between the *Etv2*-iECs, SEG-iECs, and primary endothelial samples (Fig. 5i). However, based on the number of annotated genes, SEG-iEC reprogramming was a more focused reprogramming strategy. While our results showed that there is a greater number of *Etv2*-iEC reprogramming annotated peak genes that overlap with NCEC peak genes, there are more off-target affected genes (unrelated to NCEC) when compared to SEG-iECs. GO-enrichment analysis of these annotated peak genes further highlighted the similarities and differences of *Etv2* versus *Sox17-Erg* reprogramming, with many of genes enriching for similar biological processes (Fig. 5j).

Although there are many similarities in Molecular Function, *Etv2*-iEC reprogramming has more off-target molecular functions that are not found in neonatal endothelial cells (Fig. 5k). To further contextualize how Day 3 chromatin activation correlates to transcription, we compared the differentially expressed genes from our bulk RNAseq data to the annotated SEG H3K27ac upregulated peak genes (Fig. 5l). Of the peaks that positively correlated with increased gene expression, we again observed two sets of genes, one that eventually decreased in expression by Day 7 and one that progressively increased in expression from Day 7 to 4 weeks. Included in the genes that continued to increase in expression were known endothelial marker genes (*Pecam1*, *Cdh5*, and *Vwf*) as well as cluster marker genes observed in our scRNAseq analysis (*Gja5* and *Slit2*). Since *Gja5* and *Vegfr3* were identified as genes that highlight the different mechanisms for SEG-iECs and *Etv2*-iECs, respectively, and the increase of *Gja5* expression correlates with H3K27ac activity, we then plotted the H3K27ac peaks for both *Gja5* and *Vegfr3* to compare the different conditions. While all three non-control samples showed increased peaks in *Pecam1*, *Etv2*-iECs showed the greatest peak enrichment for *Vegfr3* and SEG-iECs for *Gja5* (Fig. 5m, Supplementary Fig. 10). These differences in reprogramming mechanism and targeted fate was additionally apparent in quantitative qPCR analysis comparing known arterial, venous, and immature endothelial cell gene expression in control, *Etv2*, SEG, and native adult murine cardiac endothelial cells (Supplementary Fig. 11a, b). With the premise that the isolated native cardiac endothelial cells are heterogeneous, we observed that the SEG samples expressed similar or higher levels of arterial markers (*Dll4*, *Gja5*, and *Hey1*) compared to native cardiac endothelial cells on Day 3 and Day 7. *Etv2* samples maintained increased levels of *Vegfr3*, *Nr2f2*, and *Vwf* on Day 3 and Day 7 when compared to fibroblasts, with *Vegfr3* being expressed at higher levels than native endothelial cells on Day 3. This further confirms the more direct mechanism of SEG reprogramming by targeting a differentiated endothelial identity despite that both methods are designed to create iECs. HOMER known motif analysis further highlighted this point, with most of the known peak motifs overlapping, although SEG and *Etv2* each had their own set of unique motifs (Fig. 5n). Plotting the scRNAseq expression of the top transcription factors identified from the motif analysis revealed potential unique downstream mediators that are activated by these two reprogramming strategies with *Fli1* being associated with *Etv2* and *Ets1* for *Sox17-Erg* (Supplementary Fig. 11c). Overall, by activating this arterial-like cell fate *Sox17-Erg* is able to efficiently reprogram cardiac fibroblasts and directly transition to a mature endothelial identity without an intermediate step that is normally found in development.

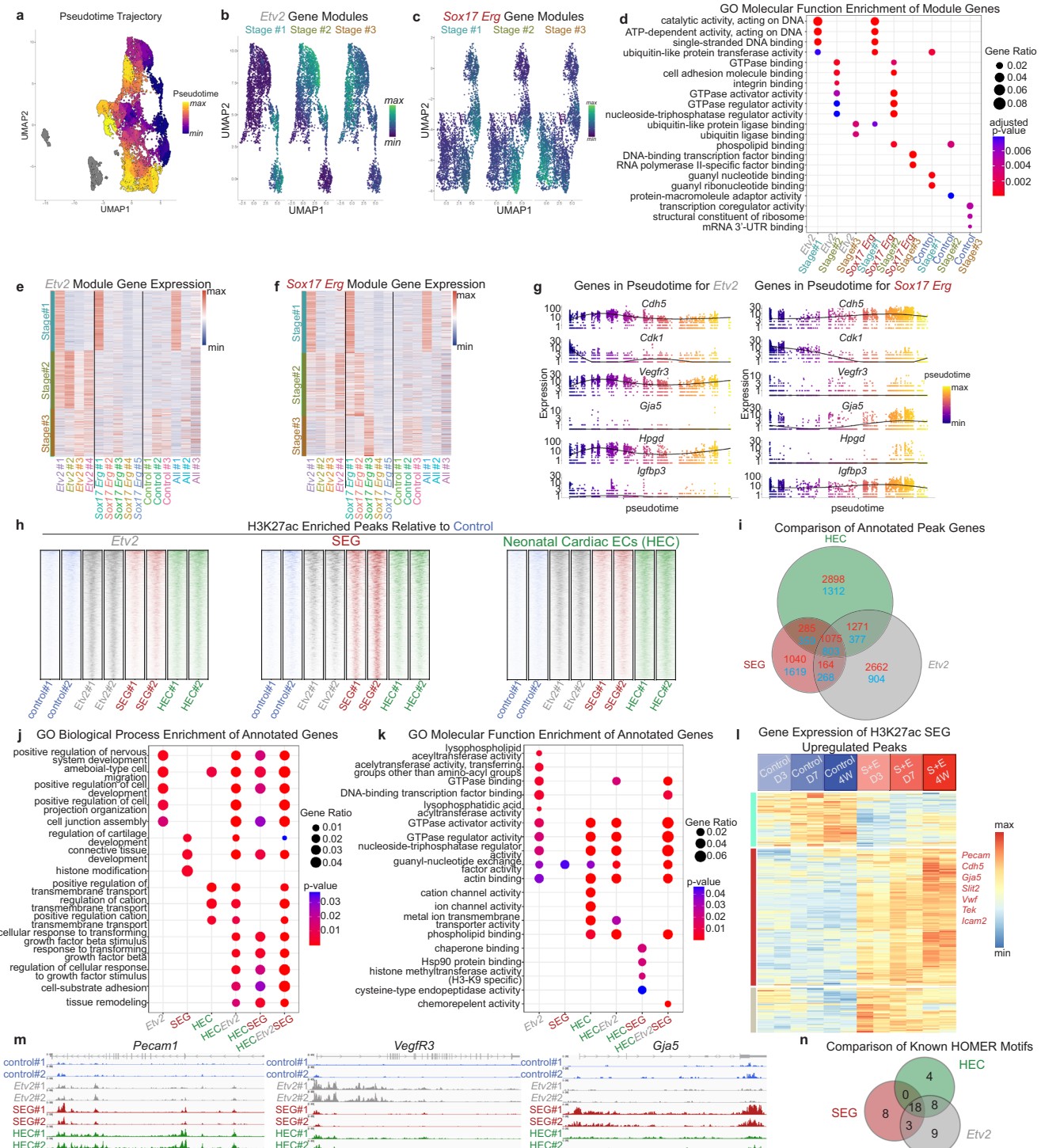

**Fig. 5 | Sox17-Erg reprogramming navigates a different route. a** UMAP of scRNAseq data annotated by pseudotime (gray cells are outside projected pseudotime). **b** UMAP of *Etv2* cluster subset of scRNAseq data annotated by *Etv2* stage gene modules. **c** UMAP of *Sox17 Erg* cluster subset of scRNAseq data annotated by *Sox17 Erg* stage gene modules. **d** Plot of GO Molecular Function enrichment of stage-specific genes identified in **b** and **c** (one-sided Fisher's exact test with Benjamini–Hochberg correction). **e** Heatmap of cluster-specific expression of *Etv2* pseudotime stage genes. **f** Heatmap of cluster-specific expression of *Sox17 Erg* pseudotime stage genes. **g** Pseudotime expression of key genes in *Etv2* and *Sox17*

*Erg* cluster subsets. **h** Heatmap of identified H3K27ac sample-specific peaks. **i** Venn diagram of annotated H3K27ac upregulated and downregulated peak genes (upregulated genes marked in red and downregulated in blue). **j** Plot of GO Biological Process enrichment of annotated genes compared in (**l**). **k** Plot of GO Molecular Function enrichment of annotated genes compared in **l** (one-sided Fisher's exact test with Benjamini–Hochberg correction). **l** Gene expression heatmap of bulk RNAseq samples of SEG H3K27ac annotated upregulated genes. **m** H3K27ac peak locations in *Pecam1*, *Vegfr3*, and *Gja5*. **n** Venn diagram of known HOMER motif of H3K27ac sample-specific peaks.

### Sox17-Erg iEC heterogeneity highlights downstream mediators

The transcriptional differences apparent in neonatal cardiac fibroblast iECs at different timepoints (Fig. 2m) led us to explore the heterogeneity in adult cardiac iECs (aiECs) using scRNAseq. Cells from control and SEG samples that were Day 3, Day 7, 2 weeks, and 4 weeks of age were multiplexed (two replicates per timepoint) and single-cell RNA sequenced, resulting in 19,939 cells for analysis (Supplementary Fig. 11a). Control and SEG samples clustered differentially with clear clusters being representative of each sample type (Fig. 6a–c, Supplementary Fig. 11b). SEG-related samples demonstrated increased expression of endothelial cell genes (*Cdh5*, *Vegfr2*, and *Pecam1*), and decreased expression of fibroblast genes (*Thy1* and *Tcf21*) as seen in the neonatal iECs (Figs. 4d–h and 6d). Of the fourteen defined clusters, control and SEG samples were linked to six clusters, each based on the sample composition of the individual clusters. All fourteen clusters had clearly defined marker genes; however only four clusters (SEG#1, SEG#3, SEG#4, and SEG#5) highly expressed the reprogramming factors, canonical endothelial marker genes (*Cdh5*, *Vegfr2*, and *Pecam1*), neonatal iEC marker genes (*Cdk1*, *Cav1*, *Slit2*), and arterial marker genes (*Gja4*, *Gja5*, *Dll4*, and *Notch1*) (Fig. 6e, f, Supplementary Fig. 12c, d). Interestingly, SEG samples showed changes in the percent composition of these clusters based on the age of the sample, with SEG#1 and SEG#4 showing steady decreases proportional to the sample age (Fig. 6c and Supplementary Fig. 12b). SEG#3 and SEG#5 both showed an increase from Day 3 to 2 weeks followed by a decrease to Day 7-like levels at 4 weeks. Gene regulatory network analysis was then performed using SCENIC to identify potential downstream transcription factor mediators of the *Sox17-Erg* reprogramming process as well as those of the resulting aiEC heterogeneity seen by the changes in cluster composition at the captured timepoints. Each cluster was identified to have unique transcription factors-linked regulons with higher average activity (Fig. 6g). The *Elk3* regulon was of particular interest due to the large number of genes in the regulon (1354 genes) and its activity being enriched in the SEG#1, which is predominately found in the Day 3 SEG samples. Further analysis revealed that *Elk3* is highly expressed in the SEG aiEC clusters with some activity in the control clusters (Fig. 6h). Sixty percent of the genes found in the *Elk3* regulon were found in the aiEC marker genes that are conserved in the four aiEC clusters, which included canonical endothelial genes (Fig. 6i). GO Biological Process enriched endothelial development and endothelial differentiation genes found in the *Elk3* regulon also overlapped with the genes with increased H3K27ac activity in both neonatal SEG and native neonatal heart endothelial cells (HEC), linking *Elk3* to the early stages of the activated *Sox17-Erg* direct reprogramming process (Fig. 6j, k). This is also indicated by in silico perturbation of *Elk3* using CellOracle, which resulted in shift vectors of not fully committed iECs (i.e. SEG cells with lower *Cdh5* and *Pecam1* expression) away from iECs with higher expression of endothelial cell markers (Supplementary Fig. 12e)[24]. Analysis of both neonatal *Etv2* and *Sox17-Erg* samples showed increased expression of *Elk3*. However, when plotted along pseudotime, the expression profiles of *Elk3* and select downstream components in *Etv2* cells showed an initial spike in expression, whereas in *Sox17-Erg* samples, a more gradual increase was apparent (Supplementary Fig. 12f–i). Despite the increase in *Elk3* expression, comparison of the SEG *Elk3* regulon genes to the upregulated H3K27ac SEG and *Etv2* genes further illustrated the similarities (*Cdh5* and *Notch1*) and differences (*Gja5*) of the early stages of these two reprogramming approaches (Supplementary Fig. 12i).

### Sox17-Erg directly reprograms cardiac fibroblasts in vivo

With one of the key advantages of direct reprogramming being the in situ conversion of the starting cell type to the targeted cell type, we then utilized the canonical fibroblast lineage tracing methodology used in iCM reprogramming to validate the conversion of adult cardiac fibroblasts into endothelial cells[2]. Adult *Tcf21*-iCre *Rosa26*-tdTomato mice were given 5 consecutive days of tamoxifen to induce the expression of tdTomato in *Tcf21*-positive cells, which are primarily fibroblasts and not endothelial cells. The mice then underwent left anterior descending coronary artery ligation surgery followed by injection of GFP or SEG retrovirus into the infarct region at least one week after the last tamoxifen injection. For both experimental conditions, the proliferating *Tcf21*-positive cells that were targeted by the injected retrovirus became GFP and *Tcf21*-tdTomato-positive. If a fibroblast converted into an induced-endothelial cell, the cell would then become GFP, *Tcf21*-tdTomato, and PECAM1 positive (Fig. 7a). The heart samples were then collected seven days post-injury, processed, imaged, and then analyzed for the percentage of retrovirus targeted *Tcf21*-positive fibroblasts that were converted into endothelial cells. As expected, the GFP retrovirus hearts resulted in no *Tcf21*-tdTomato cells expressing PECAM1 (Fig. 7b–d). SEG hearts had a statistically significant conversion of *Tcf21*-positive cells into PECAM1-positive cells (average of 35.56%) when compared to the GFP control samples (Fig. 7b, e–h and Supplementary Fig. 13), thus demonstrating the ability for *Sox17-Erg* to directly reprogram cardiac fibroblasts into endothelial cells in vivo.

## Discussion

In this study, we sought to identify a robust methodology to reprogram cardiac fibroblasts into endothelial cells and to demonstrate that these cells are able to function as endothelial cells both in vitro and in vivo. We found that *Sox17-Erg* direct reprogramming offers a highly efficient approach for reprogramming cardiac fibroblasts into induced endothelial cells. When compared to *Etv2*, this transcription factor methodology uses a more direct reprogramming mechanism that creates endothelial cells that are functional both in vivo and in vitro. Furthermore, this two-factor reprogramming cocktail can reprogram both neonatal and adult fibroblasts, highlighting its ability to reprogram cells of variable age, plasticity, and organ of origin. The ability of the *Sox17-Erg*-generated iECs to display properties that would be important for interacting with the cardiac microenvironment (such as interacting with cardiomyocytes) and to potentially impact the native cardiac vasculature at 7 days and 4 weeks post myocardial infarction illustrates the potential to generate blood vessels using organ-specific fibroblasts. However, further work needs to be done to determine how these cells directly affect the perfusion of the scar regions. This study also highlights the potential for this reprogramming to be used in situ, however, more in depth work needs to be done studying the identity of these in vivo reprogrammed cells and the impact of this reprogramming on non-endothelial cardiac repair since we did not evaluate heart function after in vivo reprogramming. Tracking the impact of in vivo iECs on cardiac function post-injury and analyzing how similar the in vivo reprogramming process is to the in vitro one through a combination of histological, cellular and multi-omics analyses would further develop this reprogramming approach.

One of the hurdles is creating organ-specific vasculature beyond cardiac vasculature. Our study illustrates the potential for the reprogrammed cells to express specific markers and functional characteristics of cardiac endothelial cells, however, further examination is required to determine how much organ-specificity is present in the reprogrammed cells (Fig. 3e and Supplementary Fig. 6c). With this in mind, direct reprogramming of organ-specific fibroblasts may be one way to help provide organ context and perhaps allow for the generation of organ-specific vasculature without the need for additional external cues. This reprogramming approach is also a proof of concept that targeting a terminally differentiated endothelial identity is possible without requiring reprogrammed cells to undergo the stepwise

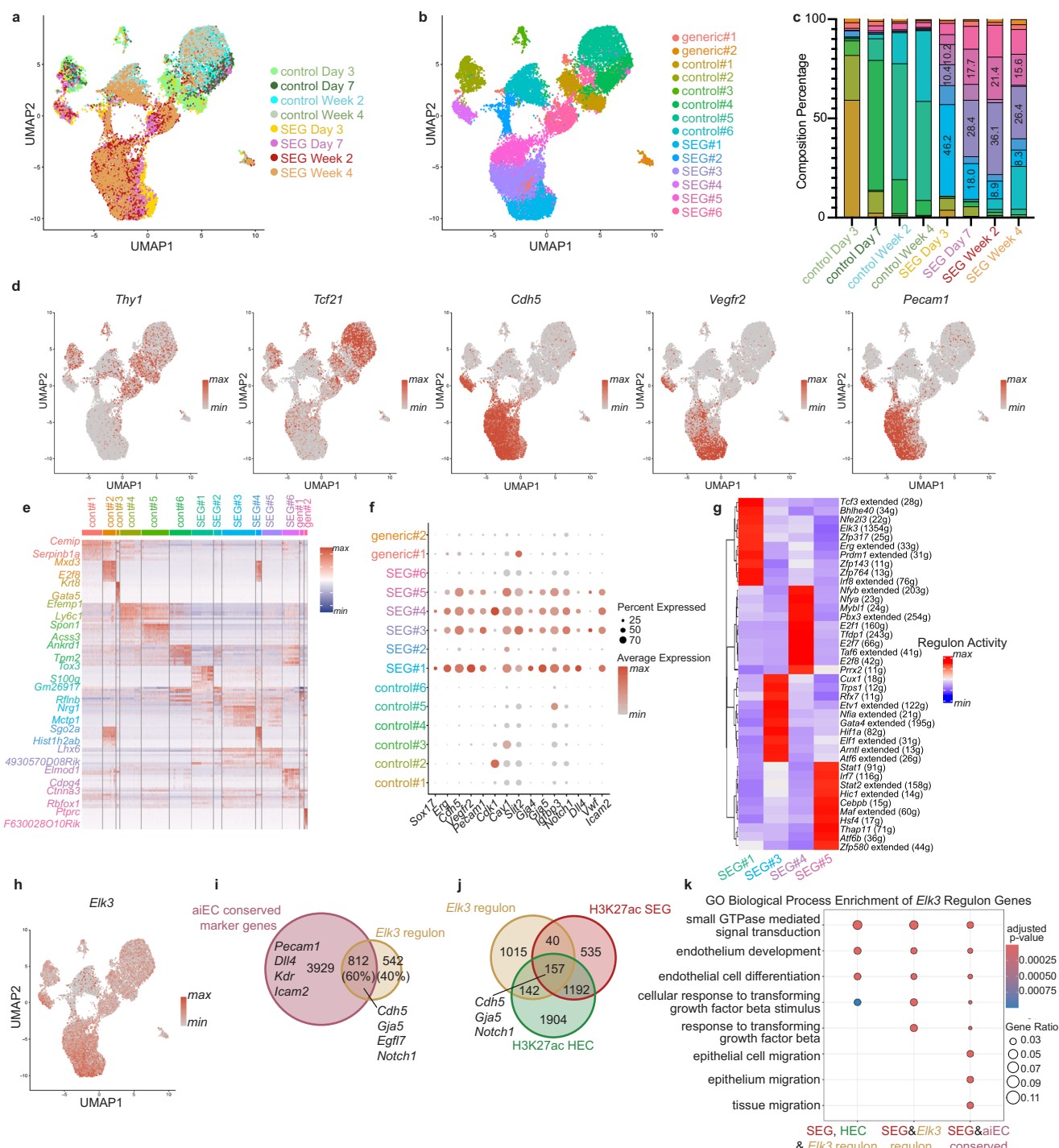

**Fig. 6 | aiEC heterogeneity changes during reprogramming. a** UMAP of aiEC scRNAseq labeled by sample type. **b** UMAP of aiEC scRNAseq labeled by cluster name. **c** Percent cluster composition of samples. Top three aiEC-related cluster percentages are labeled for SEG samples. **d**, Expression profiles of fibroblast and endothelial genes. **e** Top 10 marker gene heatmap for each cluster. **f** Cluster-specific expression of reprogramming and endothelial genes. **g** Heatmap of top 10 identified gene regulatory network regulons for selected aiEC endothelial-like clusters.

**h** Expression profile of *Elk3*. **i** Venn diagram of aiEC conserved marker genes and *Elk3* regulon genes. **j** Venn diagram of *Elk3* regulon genes, H3K27ac SEG upregulated genes in neonatal iECs and H3K27ac neonatal heart endothelial cell genes. **k** GO Biological Process Enrichment analysis of *Elk3* regulon to H3K27ac HEC and SEG, and aiEC conserved genes (one-sided Fisher's exact test with Benjamini–Hochberg correction).

progression seen in development or use *Etv2*-based strategies. However, targeting an immature cell fate would allow for opportunities to target progenitor cell types, study vascular development, and may be able to generate rarer cell-types through a more guided differentiation process. By targeting a more differentiated state, the generated cell types may be more ready to be used in a clinical setting or in situ in

damaged tissue, as seen in iCM direct reprogramming. Future studies can address what organ-specific properties are transferred during the iEC conversion, how *Sox17* and *Erg* work synergistically to induce iEC reprogramming as well as address fundamental questions on how organ-specific vessels acquire and maintain their highly specific identities.

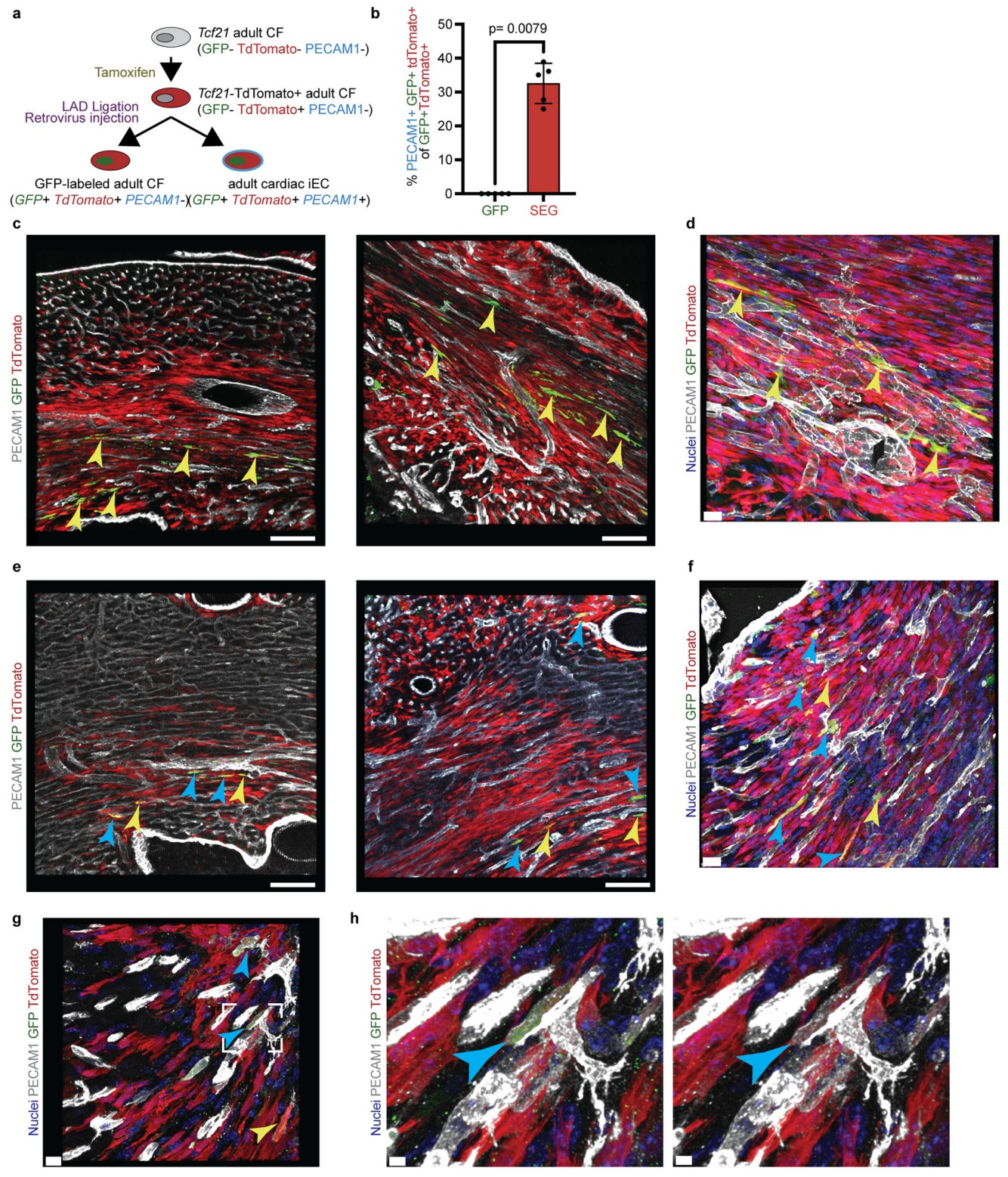

**Fig. 7 | Sox17-Erg directly reprograms cardiac fibroblasts in vivo. a** Schematic of in vivo iEC reprogramming. **b** Quantification of percent PECAM1+ GFP+ *Tcf21*-tdTomato+ of GFP+ *Tcf21*-tdTomato+ cells in Day 7 hearts (*n* = 5 animals) (two-sided Mann−Whitney test; mean with SD). Two representatives deconvoled low magnification images of Day 7 GFP control (**c**) and SEG (**e**) samples (scale bars 100 μm).

Higher magnification deconvolved images of GFP (**d**) and SEG (**f**) samples (scale bars 20 μm). **g** Additional image of in vivo iECs within region of interest (**h**) (scale bars 10 and 3 μm). Yellow arrows indicate GFP+ tdTomato+ PECAM1− cells. Blue arrows indicate GFP+ tdTomato+ PECAM1+ cells. Source data are provided as a Source Data file.

## Methods

### Mice

All mouse procedures and usage follow the ethical guidelines of and were approved by the Institutional Animal Care and Use Committee of the University of North Carolina in Chapel Hill. Neonatal and adult cells isolated and used in the described methods are isolated from *C57BL/6J* mice (Jackson Strain#000664) or *Tcf21*-CreERT2[25] bred with *Rosa26*-tdTomato (#007914). SCID mice (Charles River #236) are used in the

in vivo cell engraftment assay. Mice are housed in a facility maintained at an average of 70 °F and 50% humidity at a 12 h light cycle.

Standard 5-day tamoxifen injection protocol (IP injection of 75 mg/kg in corn oil) was used to induce *Tcf21-CreERT2* in adult *Tcf21-CreERT2* TdTomato mice (6–12 weeks old). Mice underwent surgery one to 2 weeks after the fifth injection of tamoxifen. The GFP and SEG cohorts contain mice that underwent tamoxifen injection and surgery at the same points to ensure consistency between groups.

### Myocardial infarction surgery and injection of samples

Mice were first anesthetized with a combination of ketamine (100 mg/kg) and xylazine (10 mg/kg), and then intubated and ventilated using RoVent Jr. Standard Ventilator. The heart was accessed by cutting the intercostal muscle. Once exposed, the left anterior descending artery was ligated with a 7-0 silk suture. Immediately after ligation, 20 μL of iECs suspended in sterile PBS (1 million cells per 20 μL), 20 μL of sterile PBS, or 20 μL of retrovirus (GFP or SEG) is injected in three sites surrounding the infarcted area using insulin syringes with 29 G needles (BD #305935). The chest of the animal is then sutured closed, and the animal recovers on a 37 °C heating pad.

To prepare the retrovirus for injection, the virus was harvested from platE cells, filtered, and then ultracentrifuged ($23,000 \times g$ for 2 h). Resulting pellet was then resuspended in PBS with polybrene (8 μg/mL).

Four hearts were isolated for each experimental group (PBS control group or iEC cells) seven days after surgery. Each experiment group contains hearts from 2 males and 2 females. Tissue isolated from the PBS group was used to confirm the absence of GFP signal during immunofluorescence staining. Five hearts were isolated for each experimental group (PBS control or SEG iECs) containing males and females for the 4-week study and retroviral injection.

### Lectin perfusion and analysis of myocardial infarctions

Mice were sacrificed using approved UNC IACUC methods. The heart was accessed by opening the chest cavity with scissors. The right atria was then cut, and the heart was then slowly perfused with 6 mL of PBS through a 27 G needle. The heart was then perfused with 6 mL of 10 μg/mL of DyLight 594 tomato lectin (Vector DL-1177-1) and left to incubate for 3 min. After incubation, the heart is then perfused with 8 mL of 4% paraformaldehyde and then excised into a 15 mL tube that contains 4% paraformaldehyde. Samples were sectioned at a 50-micron thickness. Three sections that were at least 100 microns apart were imaged per mouse. The myocardial scar region per section was imaged using Zeiss LSM900 using both Z-stack and tile scans. The myocardial scar region was identified based on morphology as well as the differential autofluorescence when compared to normal cardiac regions in the GFP channel. The resulting images were then flattened to a maximum intensity image and then randomly assigned a new name for blinded quantification. The resulting images were then quantified in a blinded manner using the FIJI polygon selection tool in combination with the region of interest manager to first identify the scar area and then trace the outlines of the perfused vessels. The total area of traced lectin vessels found in the section's scar area was then divided by the section's total scar area to get the percent lectin perfused scar area. For normalization of lectin perfusion to the non-scar area, representative images of the scar area for two sections per animal were collected along with a representative non-scar left ventricle region with normal lectin perfusion for each collected section. Percent lectin volume was calculated by dividing the volume of lectin measured by Imaris surfaces by the volume of tissue in the image (measured using the autofluorescence in the GFP channel).

### Imaging and analysis of day 7 Tcf21 TdTomato lineage tracing

At Day 7 post-MI surgery, mouse hearts are harvested, perfusion fixed with 4% paraformaldehyde, and sucrose dehydrated, and then cryosectioned at 50 microns. Sections are then stained for GFP and PECAM1 using the immunofluorescence protocol stated below. Three sections containing GFP-positive cells were then imaged per heart, and all GFP-positive TdTomato-positive cells found in the section were imaged as Z-stacks using Zeiss LSM900. A maximum intensity image of the Z-stack image was then generated using ImageJ for downstream analysis. Marker-positive cells were then counted using QuPath multiplex analysis based on the classification of cells using DAPI, GFP, TdTomato, and PECAM1[26].

### Neonatal cardiac fibroblast and endothelial cell isolation

In brief, hearts of neonatal mice (Day 0–Day 3) are isolated, pooled, minced, and then transferred to a 50 mL conical tube. 10 mL of 0.05% Trypsin is then added to the minced tissue, and the tube is then incubated at 37 °C for 10 min. The trypsin is then removed and 5 mL of 0.5 mg/mL Collagenase Type 2 (Worthington #LS004177) in HBSS is added to the tissue. After a 5 min incubation at 37 °C, the tube is vortexed, and the supernatant is removed and combined with 5 mL of complete media (DMEM + 10% FBS + 1% Penicillin/Strepavidin + 1% NEAA). The collagenase step is then repeated four more times, and the supernatant from each step is combined. The resulting solution of digested tissue is then passed through a 40 micron mesh sieve, and the flow through is then centrifuged for 5 minutes at $300 \times g$. If endothelial cell depletion is performed, the resulting cell pellet is resuspended in MACS buffer (0.5% BSA 2 mM EDTA in PBS), and Biotin-tagged PECAM1 antibody (BD #553371) is added to the resuspended cells (the amount of PECAM1 antibody scales with the number of isolated hearts). After a 30 min incubation at 4 °C on a rotator, the cell suspension is washed with 4 mL of MACS buffer and centrifuged for 5 min at $300 \times g$. The pellet is then resuspended in MACS buffer and Anti-biotin microbeads (Milltenyi Biotec #130-090-485) are added to the cell solution. After a 30 min incubation at 4 °C on a rotator, the cell solution is then run through a magnetic bead column (Milltenyi Biotec #130-042-401) and the flow through is collected. For endothelial cell depletion, the column is then washed with 4 mL of MACS buffer. Fibroblast isolation then follows endothelial depletion using the same methodology except for the magnetic column step, and the use of Biotin-labeled THY1 antibody (Thermo Fisher #13-0902-85). When added to the magnetic column, the initial and three 1 mL wash flow-throughs are discarded. After the wash steps, the cells still attached to the column beads are flushed with the provided plunger and then centrifuged for 5 min at $300 \times g$. The pelleted cells are then counted and resuspended in fibroblast media (DMEM + 20% FBS + 1% Penicillin/Strepavidin + 1% NEAA). If the depleted endothelial cells are used, they are flushed from the depletion column, pelleted, and then resuspended in EGM2 media (Lonza #CC-3162). All cells are plated on collagen-coated plates.

### Neonatal cardiomyocyte isolation

This procedure follows the same initial steps as fibroblast isolation. After the collagenase digestion, the digested cells are passed through a 70 micron mesh cell strainer. The filtered cells are then centrifuged at $100 \times g$ for 5 min. The supernatant is then removed, and the pellet is resuspended in complete media. The resuspended cells are then plated on a 0.1% gelatin plate and cultured for 90 min. The plated cell suspension is then removed and centrifuged for 5 minutes at $100 \times g$. The pelleted cells are then counted and then mixed with iECs at a ratio of 1 to 1.

### Adult fibroblast isolation

Adult tissues (heart, lung or skeletal muscle) are isolated from 3 to 4-month-old adult animals and washed in PBS. The washed tissue is then minced and digested in collagenase solution (1 mg/mL Collagenase Type 2 + 1 mg/mL Collagenase Type 4 (Worthington #LS0004189) in HBSS) for 30 min at 37 °C and is vortexed for 30 s every 5 min. This solution is then passed through a 40 micron mesh sieve and then

centrifuged for 10 minutes at $400 \times g$. The supernatant is then removed and the pellet is resuspended in 10 mL of ACK solution (Thermo Fisher #A1049201) and incubated at room temperature for 5 min. The cells are then centrifuged for 10 min at $400 \times g$. The pelleted cells are then resuspended in fibroblast culture media and cultured on 0.1% gelatin-collated plates for 3 h. The culture media is then removed, and the attached cells are washed with PBS twice, and new fibroblast media is added. On the next day, the media is then changed. The cells are monitored daily and then undergo MACS isolation (detailed above) when confluent, usually 4 days post-isolation.

### Retroviral constructs and infection of fibroblasts
Eight of the ten tested retrovirus constructs were cloned into the pMX-puromyocin backbone. pMX-*Tal1* (#131601) and pMX-*Klf2* (#50786) were purchased from Addgene.

platE (Cell Biolabs #RV-101) cells were used to create the appropriate retroviruses for reprogramming with Nanofect (Alstembio #NF100) or Lipofectamine 2000 (Thermo Fisher #11668019) used to transfect the platE cells using the manufacturer recommended Nanofect or Lipofectamine protocols that has been previously described[27]. Retrovirus collected on Day 2 and Day 3 post-transfection are filtered using 0.45 micron PES syringe filtered, pooled, pelleted, and resuspended in neonatal fibroblast media with polybrene (EMD Millipore #TR-1003-G). To control for batch effects, the viruses used in compared conditions are always produced on the same day using the same transfection and infection reagents.

### Immunofluorescence staining of tissue and cultured cells
All tissue is fixed in 4% paraformaldehyde and then washed with PBS. Heart tissue is then dehydrated in 15% and then 30% sucrose and then frozen in O.C.T. Compound. 10 or 50 micron sections are then sectioned from the frozen tissue using a cryostat.

Tissue sections and cultured cells are stained using the methodology using a previously published protocol. Samples are first blocked with 1% normal donkey serum in TSP (0.5% Triton X-100 and 0.1% Saponin in PBS) for one hour at 37 °C. Primary antibody solutions are then prepared using fresh blocking solution, and then the samples are incubated in primary antibody solution for one hour at 37 °C. Samples are then washed with TSP three times for 5 min each. Secondary antibody solution is then prepared using fresh blocking solution with secondary antibodies added at 1:200. Secondary antibody solution is then added, and samples are incubated for one hour at 37 °C. Samples are again washed three times for five minutes each. Samples are then stained with Hoescht to visualize nuclei and then washed one more time. PBS is then added to cultured cell samples. Tissue samples are then mounted with Prolong Diamond (#P36961) and covered with a coverslip.

Two-dimensional images were imaged using an EVOS7000 microscope. Three-dimensional images were obtained using Zeiss LSM900 at 1024 × 1024 resolution. Three-dimensional images were analyzed using Imaris and display images were deconvolved using AutoQuant.

**Antibodies.** anti-CD31 (1:100, R&D #AF3628); anti-CDH5 (1:100, R&D #AF1002); anti-GFP (1:500, Aves #GFP-1020). Donkey anti-Goat 568 (1:200, Thermo Fisher #A-11057); Donkey anti-Chicken 647(1:200, Jackson Immunoresearch (#703-605-155); Donkey anti-Chicken 488 (1:200, Jackson Immunoresearch #703-545-155). All primary antibodies were validated with controls before usage.

### Ac-LDL uptake and NO production
Ac-LDL uptake studies were performed using Cell Applications Dil-Ac-LDL Kit and protocol. Cells were treated with Dil-Ac-LDL for 4 h at 10 µg/mL in EGM2 media. Cells were washed with EGM2 media, and then nuclei were stained with Hoescht. Samples were immediately imaged using identical microscope settings during the same session and then quantified using CellProfiler using the same object identification parameters.

Nitric oxide production was detected using DAF-FM diacetate (AAT Bioquest #16298). Cells were treated with DAF-FM at a concentration of 5 mM in EGM2 media for 30 min. Cells were then rinsed with media and then incubated for an additional 15 min with Hoescht. Cells were again washed with media and then imaged using the identical microscope settings. The average pixel intensity was then measured for each image using ImageJ and then the values were normalized to the average pixel intensity of the control samples.

### TNFalpha stimulation and THP1 binding assays
Cells were treated with recombinant mouse TNFalpha at a concentration of 10 ng/mL in EGM2 media for four hours; untreated cells just received fresh EGM2 media. If collected for qPCR analysis, cells were then washed with PBS and then RNA was isolated using TRIzol extraction (Ambion #15596018). cDNA was then created from isolated RNA and then qPCR was performed on created cDNA for targeted genes.

THP1 cells were fluorescently labeled with CellTrace Far Red (Thermo Fisher#C34572) following the provided packaging instructions. After CellTrace labeling, cells were washed in THP1 culturing media (RPMI1640 + 10% FBS + 0.05 mM 2-mercaptoethanol), centrifuged at $300 \times g$ for 5 min, and then resuspended in EGM2 media. During the THP1 washing steps, TNF-alpha-treated cells are washed with EGM2 media and labeled with Hoescht dye in EGM2 media for 5 min. The labeled THP1 cells are then added to TNF-alpha-treated cells at a concentration of five hundred thousand cells per well for a 24-well plate. After a 30 min incubation, all wells are washed with EGM2 media three times. All wells are then imaged using identical microscopy settings. The number of TNF-alpha (Hoescht-positive) and THP1 (Cell-Trace-positive) cells per field was quantified using CellProfiler. The total number of THP1 and Hoescht-positive cells for each replicate was totaled for all of the collected fields and used to obtain the ratio of THP1 cells per nuclei in order to take into account potential differences of TNF-alpha stimulated cells per condition. The TNF-alpha stimulated ratios were then normalized to the appropriate non-stimulated condition.

### CellProfiler quantification of images
CellProfiler was used in the unbiased quantification of the number of cells positive for particular cell markers. When comparing different conditions or reprogramming cocktails, all samples collected for the same replicate were analyzed using identical settings that take into account signal strength and background signal to control for batch effects. All images within a replicate are acquired using identical microscope settings. A similar pipeline was used in the quantification of the number of cells positive for a particular marker. In brief, the nuclei are first identified using IdentifyPrimaryObject module. Next, the protein of interest is identified using IdentifyPrimaryObject module. The settings of this module are set by performing test images on at least three images that are randomly selected by CellProfiler. The identified nuclei are then masked by the identified marker objects using the MaskObject module, and a majority of the identified nuclei must be masked by the marker objects in order to be counted as a nucleus that is positive for the marker of interest. At least 10 adjacent images are analyzed by this pipeline per replicate of a particular test condition.

### Flow cytometry analysis
Cells are detached from the cell culture plate using Accutase (#A6964-100 mL), and an equal volume of EGM media is added to the detached cells. The cells are then centrifuged at $300 \times g$ for 5 min, and the pelleted cells are resuspended in 100 µL of FACS Buffer (0.5% BSA in PBS)

with 10 μL of APC-conjugated PECAM1 antibody (R&D FAB3628A) or 5 μL of PE-conjugated CDH5 antibody (Thermo Fisher #12-1441-82). Cells are incubated in antibody solution for 30 min at 4 °C. The samples are then washed with 1 mL of FACS buffer and centrifuged at 300 × g for 5 min (repeated three times). After the last wash, the cells are filtered through a 40 micron strainer, briefly treated with DAPI to label dead cells, and then analyzed using Thermo Fisher Atunne NxT. Collected data is then analyzed using FlowJo (Supplementary Fig. 14 for gating strategy).

### Shear stress flow experiments

Day 6 iECs were detached from well plates using accutase, centrifuged, and then were seeded in individual wells of Ibidi μ-Slide VI 0.4. On day 7, cells were washed with EGM2 media, and then their media was changed to flow media (EBM2 media with 2% FBS and 1% Penicillin/Strepavadin). Ibidi μ-Slides were attached to Masterflex Ismatec Reglo Digital Pump. The flow samples were slowly ramped up to final shear stress condition using the following protocol: 0.5 dynes/cm$^2$ (15 min), 1 dyne/cm$^2$ (15 minu), 3 dynes/cm$^2$ (15 min), 5 dynes/cm$^2$ (30 min), 10 dynes/cm$^2$ (30 min), 15 dynes/cm$^2$ (30 min), 20 dynes/cm$^2$ (1 h), 25 dynes/cm$^2$ (1 h), and 30 dynes/cm$^2$ (1 h). Flow condition samples were then subjected to 35 dynes/cm$^2$ of shear stress for 24 h using a pulsatile pump in a standard 37 °C cell incubator.

After flow conditions, samples were fixed using 4% PFA in PBS. The immunofluorescence protocol described above was then used to stain the samples for GFP and CDH5. After staining, the samples were imaged using EVOS7000 microscope. Images from each sample were then analyzed for alignment to flow using the Directionality plugin in FIJI on the GFP images (at least 9 images per sample). The Directionality of each image was calculated using the Local Gradient Orientation function, and the absolute value of the Direction of the GFP channel was recorded per image. The recorded values per sample were then averaged. The four replicates for each condition were then compared using unpaired t-test.

### qPCR of TNFalpha-stimulated samples

After 4 h TNF-alpha stimulation, cells were washed with PBS, and then the RNA was extracted from the samples using standard TRIzol extraction protocol (Thermo Fisher #115596018). cDNA was then created for each sample using SuperScript IV Vilo Mastermix (Thermo Fisher #11756050). qPCR of samples was then performed on samples in duplicate using SYBR green master mix (Thermo Fisher #4309155). The average CT value of each gene was then taken and only used if values were within a half-cycle difference. Delta CT was then calculated using beta-actin as a reference gene. The change in expression due to TNF-alpha stimulation was calculated using an unstimulated sample.

### qPCR primer sequences (5′–3′)

*Vcam1* F CTGGGAAGCTGGAACGAAGT *Vcam1* R GCCAAATTGACCGTGAC

*Sele* F ATCCTGCAGTGGTCATGGTG *Sele* R GCCAAAGGAGCAGGAGGAAT

*Icam1* F CTGGGCTTGGAGACTCAGTG *Icam1* R CCACACTCTCCGGAAACGAA

*Actb* F *CCACCATGTACCCAGGCATT* *Actb* R AGGGTGTAAAACGCAGCTCA

*Notch1* F TGGACTGTTCTGTGCATCCC *Notch1* R TGGGGATCAGAGGCCACATA

*Dll4* F GTACTCACCACTCTCCGTGC Dll4 *Dll4* R AGCTGCCACCATTTCGACAG

*Gja5* F TGAGCTCTAAACGTGGAAGGC *Gja5* R ATGGTATCGCACCGGAAGTC

*Hey1* F GTTTGG GGTTTCGGGAATGC *Hey1* R CTT CCC CAG GGA ATG TGT CC

*Vegfr3* F GACCTCCTGGTGAACGTGAG *Vegfr3* R ACG CTG GCA GA G GAG TTT AC

*Vwf* F ACAACTTGACAGCAGGTCGG *Vwf* R GCCACCTCTCACTCCTAAGC

*Nr2f2* F ACCGGGTGGTCGCTTTTA TG *Nr2f2* R GGCCTTGAGGCAGCTATACTC

*Fli1* F ATGGACGGGACTATTAAGGAGG *Fli1* R GAAGCAGTCATATCTGCCTTGG

*Kdr* F TTGCCTGGTCAAACAGCTCA *Kdr* R GCTCTGCTTCCAGGAGTGTG

### Bulk RNAseq sample collection and analysis

RNA from samples was isolated using standard TRIzol extraction protocol (Thermo Fisher #15596018). Libraries were prepared using the Kapa mRNA Stranded Kit. Samples were sequenced using paired-end sequencing at a 2 × 50 read length using an Illumina Novaseq 6000 platform by UNC HTSF core. Raw reads were demultiplexed using bcl2fastq. Samples were checked for quality control using fastqc and multiqc, and high GC content was trimmed using BBDuk. Transcript expression was then quantified using Salmon[28]. DESeq2 was used to obtain differential gene expression between control and reprogrammed samples at each timepoint[29]. Gene Ontology enrichment analysis and enrichment analysis of the differentially expressed genes to previously published datasets were performed using ClusterProfiler[30].

### Single-cell RNAseq library preparation and sequencing

Cells from two replicates for each condition were detached from cell culture plates using Accutase or 0.05% Trypsin and then centrifuged at 300 × g for 5 min at 4 °C. Replicates were then pooled and then the pooled replicates for each timepoint were multiplexed using the 3′ CellPlex Kit Set A (10x Genomics #1000261) with the Day 3, Day 7, 2 Weeks, and 4 weeks timepoints as separate samples. After the final wash step, the pooled multiplexed cells were immediately used in the Chromium Next GEM Single Cell 3′ Reagent Kit v3.1 (Dual Index) library preparation protocol (10x Genomics #1000268). The prepared gene expression and multiplexed libraries were then paired-end sequenced using NovaSeq 6000 platform by UNC HTSF core.

RNA was extracted from unused cells from scRNAseq sample preparation, and the overexpression of *Sox17*, *Erg*, and *Etv2* was independently confirmed by qPCR before samples were further analyzed.

qPCR Primer Sequences (5′–3′):

*Etv2* F cag agt cca gca ttc acc ac *Etv2* R agg aat tgc cac agc tga at

*Sox17* F gaa tcc aac cag ccc act g *Sox17* R tag gga aga ccc atc tcg gg

*Erg* F acc tca ccc ctc agt cca aa *Erg* R tgg tcg gtc cca gga tct g.

### Single cell RNAseq data analysis

The cellranger multi pipeline was used to demultiplex the sequenced samples (using the preassigned min-assignment-confidence of 0.9). The standard Seurat workflow was then used to filter (nFeatures RNA > 200, and percent.mt < 25), merge, and cluster the demultiplexed samples[31]. Once the clusters were assigned, FindAllMarkers was used to find the genes associated with each cluster with the specifications of min.pct above 25% and logfc.threshold of 0.25. ClusterProfiler was used for both GO Term Analysis and comparison to published gene sets[30]. Annotation of Tabula Muris heart data was performed by using CellID R-package[32].

Pseudotime analysis was performed by using published Monocle3 pseudotime workflow by converting the Seurat object to the cell_data_set object function found in SeuratWrappers[33]. Sample-specific trajectories were isolated using choose_cell function and gene modules describing the trajectories were identified using graph_test in combination with find_gene_modules functions (adapted from Monocle3 vignette workflow).

Cell cycle states were identified using tricycle R-package with the SeuratWrappers workflow[34]. Cell signaling for each cluster was estimated using CellChat[35]. Gene regulatory network analysis was performed using SCENIC[36]. In silico cell perturbation analysis was performed using CellOracle[24].

## H3K27ac CUT&Tag

Samples were detached from cell culture plates using Accutase, centrifuged, counted and then 100,000 cells were used in CUT&Tag V.3 protocol starting at the fresh cells step[37]. Samples were then prepared using the published CUT&Tag protocol using an H3K27ac antibody (Abcam #ab4729). Samples were then paired-end sequenced using NextSeq2000 platform by UNC HTSF core.

Data was analyzed following the published CUT&Tag data analysis protocol. The quality control statistics at each step of the analysis pipeline were routinely compared between replicates and sample types. Peaks were assigned using the SEACR package, which returned the top 0.01 fraction of stringent peaks per sample. Differential peaks were assigned by comparing the assigned peaks of each condition (*Etv2*, SEG, or primary endothelial cells) to the control condition using DESeq2 R-package[29]. Deeptools were used to generate peak heatmap for each sample[38]. HOMER package was used for transcription factor motif analysis of the differentially expressed peaks per sample type (*p*-value cut-off of 0.01 for Venn diagram comparison)[39]. The differentially expressed peaks were annotated using ChIPseeker R-packge, and ClusterProfiler was used for enrichment analysis of the annotated genes[30,40].

## Statistics and reproducibility

GraphPad Prism was used to perform all of the statistical analyses. Comparisons were evaluated using two-sided *t*-tests, the Mann–Whitney test (for non-normal data), or One-way ANOVA with Tukey's multiple comparison tests with a *p*-value of 0.05 being the minimum cut-off for statistical significance. For the screening data, ratio-paired *t*-tests were used to take virus titer variability and differences in starting samples into account. For all other *t*-tests, an unpaired *t*-test with Welch's correction was used. No statistical method was used to predetermine the sample size. No data were excluded from the analyses. Sample blinding was used for the analysis of in vivo data. When possible, all quantifications of immunofluorescence data were automated using CellProfiler or QuPath.

## Reporting summary

Further information on research design is available in the Nature Portfolio Reporting Summary linked to this article.

## Data availability

The data generated in this study have been deposited in the NIH GEO database under accession code GSE218418. The embryonic day 12.5 data used in this study are available in the Github database (https://github.com/gmstanle/coronary-progenitor-scRNAseq). The adult murine heart scRNAseq data used in this study are available in the Figshare database under Tabula Muris [https://figshare.com/projects/Tabula_Muris_Transcriptomic_characterization_of_20_organs_and_tissues_from_Mus_musculus_at_single_cell_resolution/27733]. Source data are provided with this paper.

## Code availability

Bioinformatics code used to run analyses used in this manuscript can be made available upon request.

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

## Acknowledgements

We would also like to acknowledge UNC-Chapel Hill's Flow Cytometry Core (NIH/NCI P30CA016086), High-Throughput Sequencing Facility (NIH/NCI P30CA016086 and NIH/NIEHS P30ES010126), and Microscopy Services Laboratory (funded in part by P30 CA016086 Cancer Center Core Support Grant to the UNC Lineberger Comprehensive Cancer Center). We would like to thank Dr. Esther Gu Farber for helping edit this manuscript. This work was supported by AHA Postdoctoral Fellowship no. 825942 to G.F.; AHA Postdoctoral Fellowship 927906 to H.F.W.; 5F30HL154659 to B.K.; AHA20EIA35310348 NIH/NHLBI R35HL155656 to L.Q., NIH/NHLBI R01HL139976 and R01HL139880, and AHA 20EIA35320128 to J.L.

## Author contributions

G.F. and L.Q. designed the study. G.F., Y.D., Q.W., M.R., M.D., and K.B. performed the experiments. G.F., H.F.W., B.K., and Y.X. analyzed the data. G.F. and L.Q. wrote the manuscript. W.J.P., J.L., and L.Q. supervised the work.

## Competing interests

The authors declare no competing interests.
