## [Peer Review File · Nature Communications]

REVIEWER COMMENTS

Reviewer #1 (Remarks to the Author):

GENERAL COMMENTS

Farber and colleagues present a manuscript examining a new strategy to reprogram fibroblasts to endothelial cells using forced expression of Sox17 and Erg. In particular, the authors focus on the cardiac system using cardiac fibroblasts from mice to devise a strategy to directly reprogram to arterial-like endothelial cells, a desired cell type for repair of ischemic cardiac tissue. The authors systematically test forced expression of a variety of candidate factors based on relevant gene expression studies. Ultimately, they identify a combination of Sox17-Erg as the most efficient to generate induced arterial-like endothelial cells from murine cardiac fibroblasts. They suggest this approach is more efficient than the previously described Etv2-based reprogramming which produces a more immature endothelial phenotype.

The manuscript has a number of strengths. It is clearly presented and written. The authors are systematic in their evaluation of different factor combinations to achieve the optimal reprogramming cocktail. Both gene expression and functional studies support the iEC cell reprogramming. The extensive transcriptional analysis with bulk RNA-seq and single cell RNA-seq provide clear insights into reprogramming trajectories and cell fates comparing Etv2 and Sox17-Erg based strategies. It is encouraging that the cells appropriately engraft into apparent vascular structures in the post-MI heart.

However, there are some limitations to the current presentation:

1) What is the absolute reprogramming efficiency for data in Figure 1 g-j? Only normalized data are presented, and it would help the readers put the findings in context if the absolute % Pecam-positive cells is provided in the text or legend. Figure 3bc, same question regarding absolute percentage of starting cells reprogrammed.

2) The flow-based alignment data is not convincing as presented in Fig. 3i. The choice of the bins used for statistical comparisons seems somewhat arbitrary in order to show small but significant differences. It is not clear from the methods how this was done as the methods state, "The collected data from the ten images were then averaged per sample, the corresponding positive and negative bins combined, the data above 45-degrees combined, and the data below 45-degrees combined (excluding 0-degrees bin)." How does this strategy relate to what is presented as 0+22degrees, etc.? Excluding 0-degrees bin is strange as $0 \pm 10^\circ$ seems the most relevant of the 9 bins to look for alignment. The authors need to convince the readers that the change in alignment is a robust finding.

3) 'Notably, we observed that iECs were able to both attach to continuous vessels and form bifurcations...' overstates the data. The fact that GFP positive cells occasionally are seen at points of bifurcation (Fig.3l) does not mean they formed the bifurcations. This structure may have preceded the transplant and iECs engrafted at the bifurcation. A more cautious statement of the findings is suggested.

4) Figure 4p, shows significant difference between control and SEG, but not Etv2 and SEG. Thus, the authors conclusion that reprogramming efficiency is greater for SEG than Etv2 is not directly supported as presented.

5) The results beg the question as to whether this reprogramming strategy is effective for other fibroblasts, e.g. dermal, or cells types from blood, that would make it more readily translatable. Have the authors tested other starting cell sources?

MINOR COMMENTS

1) For Fig. 1b, the authors second column is entitled, 'Found in reprogramming?' This is odd wording for what they are presenting, suggest considering 'Used in reprogramming?'"

2) Fig. 4b, the color scheme makes it difficult to discriminate between different populations. The subtle contrast in colors between ETV2 daye, Etv2 day 7, Control day 3, and Control Day 7 reduces readability.

Reviewer #2 (Remarks to the Author):

In this manuscript, Faber et al. showed a novel direct reprogramming method to induce endothelial cells from mouse cardiac fibroblasts. The authors newly found a gene set that induced conversion of fibroblasts to endothelial cells, Sox17 and Erg. Endothelial cells induced this method showed various different features from endothelial cells converted with Etv2, which were previously reported. It is significant that new core genes for endothelial cell conversion have been found. Nevertheless, many arguments of the authors have not been properly supported with valid evidences. The authors should add more valid experiments and/or have more logical arguments.

Specific comments are as follows:

1. In Fig. 2, the authors performed endothelial cell depletion with a sequential MACS procedure. It is unclear how was the effect of this process. The authors should show, for example, FACS analysis for Thy1 and PECAM1 before and after the depletion process.

2. In Fig. 2e, 3b, and 3c, few but some PECAM1+ (or Cdh5+) cells appeared even in the control conditions. How do the authors consider these populations? Even in the control condition, some cells can be converted into endothelial cells? These cells should be a contamination of primary endothelial cells even after the endothelial cell depletion? If the latter, still possibility that Sox17 and Erg enhance endothelial cell proliferation but not conversion should not be completely excluded. Marking for original fibroblasts should be done more strictly, especially if the authors' main argument is "conversion" of fibroblasts to ECs.

3. Quantification of endothelial cell appearance seemed to be done mainly with immunostaining (Fig. 2e, 3b, 3c). As the EC appearance look different between FACS and immunostaining (i.e. Fig. 2b and 2d), such quantification should be done also with FACS analysis. PECAM1/Cdh5 double staining and additionally Thy1 would be essential.

4. What cell types are PECAM1-negative cells after the Sox17/Erg gene transduction? Still be fibroblasts? Or becoming other cell types (partially reprogrammed)?

Several critical authors' arguments are not fully supported with valid data;

5. Creation of "cardiac" endothelial cells: the authors mentioned "Sox17-Erg-generated iECs display cardiac-specific properties (such as interacting with cardiomyocytes)". The data for this notion is based on Fig. 3f. This data shows just one example of non-clear results. Not compared with other ECs, nor quantitatively evaluated. The authors' argument is overinterpretation only from such preliminary results.

6. Engraftment ability to the heart: the authors just examined the existence of injected Sox17-Erg-generated ECs only at 7 days after the transplantation with no clear lumen formation. Again, not compared with other ECs, nor quantitatively evaluated. Engraftment ability should be estimated with longer-term, such as one month or three months. The bind of transplanted iECs to naïve ECs is unclear. Staining of functional vessels with the injection of lectin-GFP to murine vessels would be help for the judgement.

7. Induction of mature ECs: it is unclear which data clearly support the authors' argument. VEGFR3 expression does not determine the maturation level of ECs. Other features that the authors examined,

such as Nitric oxide production (Fig. 2g), alignment to flow (Fig. 3i) and so on, showed very subtle difference that looks insufficient to conclude the EC maturation.

8. In Fig. 4n, the authors showed that cell cycle status was different at day 3 after Sox17-Erg transduction. On the other hand, in Fig. 5d, Sox17-Erg- and Etv2- induced ECs showed similar gene ontology in stage 1 and 2. It is rather confusing. Actually, when does the difference in Sox17-Erg- and Etv2- induced ECs start to appear?

9. It is somewhat acceptable that Sox17-Erg-generated ECs are arterial-like, and Etv2-induced ones are venous or lymphatic-like. But still the evidence is scarce. Not just several markers (i.e. Gja5, VEGFR3 etc.) but more comprehensive analysis comparing with endogenous primary ECs would be required for more convincing conclusion.

10. Endogenous Etv2 is expressed in a progenitor stage of ECs but not in ECs. Continuous expression of Etv2 after the conversion to ECs would be considered to hinder EC maturation. So, it would be likely that the maturation status of Sox17-Erg-generated ECs is relatively higher than that of Etv2-induced ECs. But that would not mean that Sox17-Erg-generated ECs are mature ECs.

Reviewer #3 (Remarks to the Author):

The manuscript has generated extensive biological data sets, which are both innovative and significant. I have the following suggestions/questions for improving the manuscript:

Specifics:

1) The claim of that the iECs display a significantly better functional property of endothelial cells, a better interact with cardiomyocytes, and better engraft in mouse model of AMI, are significant ones. You need to demonstrate these findings, and describe each of them quantitatively in the Abstract.

2) Likewise, the claim that iECs are more arterial in nature (compared to ETV2 derived?) need quantitative data to support.

3) The conclusion of that iECs are more mature endothelial state than the traditional Etv2 reprogramming derived ECs also need the quantitative data to support, which need to be described in

the abstract. Why it is important? The pros and cons of mature vs immature ECs should be discussed in Discussion.

Reviewer #4 (Remarks to the Author):

This study is a logical extension from the Qian laboratory identifying novel factors for endothelial reprogramming. Collectively, the manuscript demonstrates that Sox17-Erg can efficiently reprogram cardiac fibroblasts into endothelial cells having an arterial-like identity which are reportedly more mature endothelial population compared to the traditional Etv2-mediated reprogramming generated endothelial cells. The manuscript is well written and the figures are of high quality. The major criticism is that these studies lack new mechanistic insights and is primarily a descriptive study. The *in vivo* studies add very limited new insights—the study that should have been performed is a fate mapping approach with the delivery of Sox17-Erg into the heart following injury that demonstrates a functional improvement in cardiac function. Other minor issues are outlined below in no particular order.

- 1) The positive-negative selection for fibroblasts should include a negative selection for the endothelial progenitors such as Flk1 and Tie2. CD31 is an intermediate endothelial marker.
- 2) In figure 1 g-h, the authors should plot the actual %-Pecam-positive cells rather than normalization to the 10 factors (used to initiate the assay). This will help gauge the overall efficiency of the reprogramming. Moreover the quantification should be performed using both FACS and immunohistochemistry. For example, panel 1n does not correlate with any of the quantitative data.
- 3) As outlined in Figure 2, the authors should provide a time course analysis of the reprogramming studies that were performed up to 4 weeks. Quantification of the endothelial reprogrammed cells over the specified time course will help determine whether the endothelial cells continue to increase or reach a plateau after a certain time period during the reprogramming process.
- 4) The GO Biological Processes provided in Figure 2 should be placed in the supplemental data section.
- 5) Although the study has undertaken several functional assays for the reprogrammed endothelial cells, the authors should include tube formation assays.
- 6) The authors report a difference in the efficiency of the endothelial reprogramming while using the polycistronic constructs. Did they look for an actual difference in the expression of Sox17 and ERG depending on their position in the cassette?
- 7) As outlined above, the *in vivo* studies are really a major detraction for this manuscript. While the authors have performed myocardial infarction injuries and have injected the iECs to demonstrate engraftment, did they perform any assays to assess whether this engraftment leads to an improvement of the cardiac function in the hearts injected with iECs vs. control infarcted animals.

8) Did the authors evaluate the potential of only the Pecam-positive population compared to the entire population of fibroblasts undergoing reprogramming?

9) Regarding the single cell analyses performed, were there any insights regarding the molecular programs that were uniquely governed by Sox17-Erg vs. ETV2? Does Sox17-Erg induce ETV2 expression or Flk1 expression? Is Sox17-Erg a pioneer factor complex?

10) To distinguish the mechanistic differences between the different networks governed by Etv2 vs Sox17-Erg, CHIP-Seq or Cut & Run in combination with ATAC-Seq analysis and the scRNASeq dataset will allow the authors to identify the different gene sets activated by the respective method of reprogramming in the cardiac fibroblasts.

Thank you for the excitement, insightful comments and constructive criticisms from all editors and reviewers involved in this manuscript. We executed new experiments, modified existing figures, and reperformed experiments from the original manuscript in response to reviewer comments. Overall, the added experimental data focused on the following areas: Flow cytometry of dual-labeled (Cdh5 and Pecam1) iEC samples and Tcf21-tdTomato lineage traced samples, four-week in vivo studies with echocardiography and lectin perfusion analysis, endothelial functional studies with native endothelial cells, and comparative endothelial marker expression of Etv2, Sox17-Erg, and adult murine cardiac endothelial cells.

Please see below for more details on how we addressed specific comments. To aid in the revision process, we have also included abridged figures that are specific to certain comments to highlight the improvements to this manuscript. These rebuttal letter figures are in addition to our revised manuscript figures, and we also indicated where this new data can be found in the main manuscript.

We would like to again thank both the reviewers and editors for their time and effort to help improve this manuscript. We believe that we addressed all of the reviewer comments to the best of our abilities and that this manuscript is greatly improved as a result of these efforts.

Changes to Address Specific Comments:

Reviewer #1

1) What is the absolute reprogramming efficiency for data in Figure 1 g-j? Only normalized data are presented, and it would help the readers put the findings in context if the absolute % Pecam-positive cells is provided in the text or legend. Figure 3bc, same question regarding absolute percentage of starting cells reprogrammed.

We changed the graphs in **Figure 1g-j** to reflect absolute percentage of reprogramming efficiency. For **Figure 3b,c**, we do show the absolute percentage for conversion efficiency (**Figure 3b**) and Pecam-positive cells (**Figure 3c**) in our original figure, however our initial wording of "...relative to..." was misleading so we have change that to "Percent Pecam+GFP+ of GFP+ cells" or "Percent Pecam+GFP+ of total nuclei." We also included the most pertinent average percent Pecam1 positive cells in the main text for both the screen and polycistronic construct comparison.

2) The flow-based alignment data is not convincing as presented in Fig. 3i. The choice of the bins used for statistical comparisons seems somewhat arbitrary in order to show small but significant differences. It is not clear from the methods how this was done as the methods state, "The collected data from the ten images were then averaged per sample, the corresponding positive and negative bins combined, the data above 45-degrees combined, and the data below 45-degrees combined (excluding 0-degrees bin)." How does this strategy relate to what is presented as 0+22degrees, etc.? Excluding 0-degrees bin is strange as $0 \pm 10^\circ$ seems the most relevant of the 9 bins to look for alignment. The authors need to convince the readers that the change in alignment is a robust finding.

We appreciate this comment and have simplified our approach for measuring the response to shear stress to illustrate our results. Instead of comparing the angles of binned pixels, we measured the overall directionality of individual images per condition using the FIJI Directionality

plugin. We then performed a t-test of the average angle of directionality comparing the flow to static conditions, and show that the flow condition angle (mean of 33.78 degrees) is less than the static condition (mean of 47.77 degrees) (**Revision Figure 1** or **Figure 3i**). This reduction in angle (or approach towards zero) shows that the cells undergoing flow are aligning towards the direction of flow, which is set at zero degrees in our analysis. We also modified the text to “began to align with...the direction of flow” to more accurately reflect the data and heterogeneity of the cells used in the shear stress experiments.

Revision Figure 1: Updated Analysis of Alignment to Shear Stress (* p -value < 0.05)

3) ‘Notably, we observed that iECs were able to both attach to continuous vessels and form bifurcations...’ overstates the data. The fact that GFP positive cells occasionally are seen at points of bifurcation (Fig.3i) does not mean they formed the bifurcations. This structure may have preceded the transplant and iECs engrafted at the bifurcation. A more cautious statement of the findings is suggested.

We have toned down the wording and interpretation of the Day 7 in vivo data to only reflect the presence of the GFP+ iECs attaching to existing endothelial cells.

4) Figure 4p, shows significant difference between control and SEG, but not Etv2 and SEG. Thus, the authors conclusion that reprogramming efficiency is greater for SEG than Etv2 is not directly supported as presented.

Day 7 Adult Cardiac Fibroblast Reprogramming

Revision Figure 2: Updated comparison of Day 7 adult iEC quantification

We have updated **Figure 4p** with a direct t-test comparing SEG to Etv2 reprogramming which illustrates that SEG reprogramming results in a greater number of Pecam+ cells when compared to Etv2 (**Revision Figure 2** and **Figure 4p**).

5) The results beg the question as to whether this reprogramming strategy is effective for other fibroblasts, e.g. dermal, or cells types from blood, that would make it more readily translatable. Have the authors tested other starting cell sources?

Thank you for this comment. We have begun to test SEG reprogramming on other organ-specific fibroblasts and have planned future experiments to explore how efficient SEG reprogramming is in other fibroblast types. For this manuscript, we have included new data showing that SEG can reprogram both adult murine lung and skeletal muscle fibroblasts into Pecam-positive cells (**Supplementary Figure 8**). We have incorporated this data into the manuscript to further show that SEG shows promise in reprogramming other adult fibroblasts as seen by the greater number of Pecam-positive cell present by Day 7 (**Revision Figure 3** or **Supplementary Figure 8**).

Revision Figure 3: Quantification of Day 7 Lung and Skeletal Muscle Fibroblast iEC Reprogramming

MINOR COMMENTS

1) For Fig. 1b, the authors second column is entitled, 'Found in reprogramming?' This is odd wording for what they are presenting, suggest considering 'Used in reprogramming?'

We have changed the second column title to "Used in reprogramming."

2) Fig. 4b, the color scheme makes it difficult to discriminate between different populations. The subtle contrast in colors between ETV2 day 2, ETV2 day 7, Control day 3, and Control Day 7 reduces readability.

We appreciate this comment, and have changed all ETV2-related data to gray (day 3) or black (day 7) in all figures found in this manuscript.

Reviewer #2

1. In Fig. 2, the authors performed endothelial cell depletion with a sequential MACS procedure. It is unclear how was the effect of this process. The authors should show, for example, FACS analysis for Thy1 and PECAM1 before and after the depletion process.

To address this point, we have performed Pecam Thy1 flow cytometry comparing isolated fibroblasts that underwent Pecam1 depletion and fibroblasts that did not. These fibroblasts were isolated, then cultured for seven days in culture (to take into account the affect that the culture conditions may have on cell identity), and finally analyzed by flow cytometry (**Revision Figure 4 and Supplementary Figure 2**). We did see a downward trend in both the percent Pecam-positive and Pecam-positive Thy1-negative after Pecam depletion when compared to no depletion, however, the results were not statistically significant (**Revision Figure 4** for quantification of flow cytometry data).

2. In Fig. 2e, 3b, and 3c, few but some PECAM1+ (or Cdh5+) cells appeared even in the control conditions. How do the authors consider these populations? Even in the control condition, some cells can be converted into endothelial cells? These cells should be a contamination of primary endothelial cells even after the endothelial cell depletion? If the latter, still possibility that Sox17 and Erg enhance endothelial cell proliferation but not conversion should not be completely excluded. Marking for original fibroblasts should be done more strictly, especially if the authors' main argument is "conversion" of fibroblasts to ECs.

This is an important point that we have addressed by including flow cytometry data that analyzes cardiac fibroblasts isolated from adult Tcf21-CreERT2 ROSA26-tdTomato lineage traced mice. As our newly included depletion data shows above (**Supplementary Figure 2 and Revision Figure 3**), using MACS to isolate Thy1 positive cells does result in very minimal but present native endothelial cell contamination. To prove that we have fibroblast to endothelial cell direct reprogramming, we labeled Tcf21-positive cells (a marker of a population of native cardiac fibroblasts) in adult mice using the tamoxifen inducible Tcf21-Cre system in conjugation

with ROSA26-tdTomato. The cardiac fibroblasts from these mice were then isolated using our adult cardiac fibroblast isolation method which includes Thy1+ Pecam- MACS isolation. These cells then underwent direct reprogramming and then analyzed by flow cytometry at the Day 7 stage of reprogramming. If we have conversion, we expect to see cells that are Pecam+ Tcf21 tdTomato+. Our results show that SEG results in a statistically significant increase in the number of Tcf21-tdTomato+ Pecam+ cells when compared to our negative control signifying that this is a conversion process of the Tcf21-tdTomato+ Pecam- fibroblast population to Tcf21-tdTomato+ Pecam+ cells (**Figure 4r,s** and **Revision Figure 5**).

Revision Figure 5: Flow Cytometry of Day 7 Tcf21-tdTomato labeled iECs. a) representative flow cytometry data. b) quantification of percent Tcf21-tdTomato+ Pecam1+ cells. (**** p-value < 0.0001)

3. Quantification of endothelial cell appearance seemed to be done mainly with immunostaining (Fig. 2e, 3b, 3c). As the EC appearance look different between FACS and immunostaining (i.e. Fig. 2b and 2d), such quantification should be done also with FACS analysis. PECAM1/Cdh5 double staining and additionally Thy1 would be essential.

We performed and added Cdh5 Pecam double staining flow cytometry data of one week, two week and four week samples (**Figure 2b-e**, **Revision Figure 6a** for representative Day 7 data, and **Revision Figure 6b** for graph summary of the different timepoints), as well as Pecam Tcf21-tdTomato staining of one week samples (**Figure 4r,s** and **Revision Figure 5**).

Revision Figure 6: Flow Cytometry Cdh5 Pecam Dual Staining. a) Representative Day 7 data of control, Etv2, and Sox17-Erg conditions. b) Graph of % Pecam+ Cdh5+ of Day 7, 2 weeks, and 4 weeks samples (** p-value < 0.01, *** p-value < 0.001)

4. What cell types are PECAM1-negative cells after the Sox17/Erg gene transduction? Still be fibroblasts? Or becoming other cell types (partially reprogrammed)?

This is an interesting question that can be addressed on the surface level with our scRNAseq annotation data (**Figure 4m**), however, more in depth and concrete answers would require experiments out of the scope of this initial paper focused on introducing Sox17 and Erg reprogramming as one method to generate induced endothelial cells. Our annotation data suggests that by Day 7 Sox17-Erg reprogramming primarily results in a combination of endothelial cells, cardiac muscle cells, and myofibroblasts. This combination of cell types is similar to what is found in Etv2 reprogramming in our scRNAseq data. The control condition, on the other hand, remains mostly comprised of fibroblasts and cardiac muscle cells, with some portion of myofibroblasts. Overall, this data indicates that it is likely that the unsuccessfully reprogrammed cells revert back to a fibroblast-like identity; which has previously been shown to occur in induced-cardiomyocyte reprogramming¹. However, it is hard to parse a full reversion compared to partial reprogramming since the cells may have some level of Sox17 and Erg expression not native to fibroblast identity.

5. Creation of “cardiac” endothelial cells: the authors mentioned “Sox17-Erg-generated iECs display cardiac-specific properties (such as interacting with cardiomyocytes)”. The data for this notion is based on Fig. 3f. This data shows just one example of non-clear results. Not compared with other ECs, nor quantitatively evaluated. The authors’ argument is overinterpretation only from such preliminary results.

We appreciate this comment and have now included a co-culture experiment in which the SEG iECs are now co-cultured with cardiomyocytes for five days, and the cardiomyocyte Gja1 gap junction interaction with SEG iECs is compared to a negative control (GFP+ fibroblasts) and isolated native neonatal cardiac endothelial cells (Pecam+ cells). We then quantified the interaction between cardiomyocytes and the other culture cells by measuring the percent of cardiomyocytes that have overlapping Gja1+ junctions with the other cell type of interest (**Figure 3e,f and Revision Figure 7**).

Revision Figure 7: Updated Cardiomyocyte iEC interaction using Gja1 quantification. a) representative images for control fibroblasts and SEG iECs (GFP for fibroblast or iEC, Pecam1 for native endothelial cell, cTNT for cardiomyocyte, Gja1 for Cx43 gap junctions).

We have also added a plot analyzing the expression of a curated set of cardiac endothelial marker genes found in the literature in the scRNAseq cell clusters to illustrate that specific Sox17-Erg (SE) clusters express more cardiac-like genes (Fbln1, Rftn1, Lamb1, and Myadm) than Etv2 clusters² (**Extended Data Fig. 6c and Revision Figure 8**). While both of these results show promising cardiac-specific traits in the iECs, we have also toned down the wording of the discussion section to reflect the caveats of these experiments and of labeling these iECs as cardiac-specific.

6. Engraftment ability to the heart: the authors just examined the existence of injected Sox17-Erg-generated ECs only at 7 days after the transplantation with no clear lumen formation. Again, not compared with other ECs, nor quantitatively evaluated. Engraftment ability should be estimated with longer-term, such as one month or three months. The bind of transplanted iECs to naïve ECs is unclear. Staining of functional vessels with the injection of lectin-GFP to murine vessels would be help for the judgement.

To further address the functional implications of injecting iECs at the time of infarct, we have added new data that analyzes perfusion of the myocardial scar as well as echocardiography of the mice pre-infarct and four weeks post-infarct. We injected mice with lectin at four weeks post-infarct surgery through the left ventricle to determine whether iEC injection improved the perfusion of the scar area. Quantification of the area of lectin perfused vessels relative to normal non-scar area showed significant improvement in the iEC injected samples when compared to PBS samples (**Figure 3n and Revision Figure 9b**). We have also toned down the wording of engraftment for the Day 7 experiments to reflect the caveats of our data and analysis. As our new data four-week data indicates, we do not see a difference in cardiac function post-myocardial infarction when comparing iEC to PBS control samples (**Figure 3m and Revision Figure 9a**). This is not a surprise given that we are injecting new cells into the scar area, not reprogramming the in-situ fibroblasts, and it may take a longer time course for cardiac function to improve. Additionally, we envision these iECs as being used in combination with iCMs in improving heart function since iECs have a more indirect influence on heart function than cardiomyocytes.

We limited the scope and interpretations of these experiments due to technical challenges of these experiments (i.e. high mortality of the induced infarct, cells potentially being washed out during the four week waiting period, and, to our knowledge, the relative novelty of injecting endothelial cells into myocardial scar) and to keep this manuscript focused on the initial characterization of this reprogramming methodology. However, our four week data does show encouraging results that the perfusion of the scar areas are improved in our iEC samples compared to PBS controls which we hope to pursue in future studies.

7. Induction of mature ECs: it is unclear which data clearly support the authors' argument. VEGFR3 expression does not determine the maturation level of ECs. Other features that the authors examined, such as Nitric oxide production (Fig. 2g), alignment to flow (Fig. 3i) and so on, showed very subtle difference that looks insufficient to conclude the EC maturation.

Revision Figure 8: scRNAseq dot plot of Heart-specific endothelial cell marker genes. *Fln1*, *Rfn1*, *Lamb1*, and *Myadn* are boxed to denote higher expression in SE clusters compared to *Etv2* clusters.

Revision Figure 9: a) Fractional shortening pre-infarct (Initial) and post-infarct for PBS and iEC samples. b) MI scar perfusion area relative to non-scar area

We have performed two sets of experiments to more directly address endothelial cell maturation status. First, we have reperformed our Ac-LDL and NO production experiments to include native heart endothelial cells to allow for direct comparison of control, Sox17-Erg and native heart endothelial cells (**Figure 2h,2i**). While the Sox17-Erg iECs may not have identical levels to heart endothelial cells in both cases, we do see that Ac-LDL+ cells do increase to more native endothelial-like levels (**Figure 2h**) and an elevation in NO Production compared to control samples (**Figure 2i**). These differences are likely due to direct reprogramming heterogeneity (samples contain fibroblasts and endothelial cells) and native endothelial cell heterogeneity. Second, to further illustrate the maturation/identity of the generated iECs, we have included qPCR data comparing control adult cardiac fibroblasts, Etv2 adult iECs, SEG adult iECs, and native adult murine heart endothelial cells (**Extended Data Fig. 11**). Our provided qPCR data provides insight into SEG iECs expressing more arterial-like markers (i.e. Dll4, Gja5, Hey1) than control cells while only lowly expressing immature (Vegfr3) or venous markers (Nr2f2 or Vwf) by Day 7. The SEG iEC expression levels of the arterial markers are either similar or greater than the levels expressed by the isolated native adult cardiac endothelial cells suggesting that the SEG iECs are more arterial-like due to the expected heterogeneity of the native endothelial cell samples. Overall, we interpret the combination of expressing arterial markers at levels similar to mature native endothelial cells and a portion of the heterogeneous iEC population displaying mature EC functional properties as Sox17-Erg generating mature arterial-like induced endothelial cells. To provide further context for the audience, we have added to our Discussion section to reflect the intricacies of defining these iECs as mature and comparing the pros and cons of creating mature and immature endothelial cells.

8. In Fig. 4n, the authors showed that cell cycle status was different at day 3 after Sox17-Erg transduction. On the other hand, in Fig. 5d, Sox17-Erg- and Etv2- induced ECs showed similar gene ontology in stage 1 and 2. It is rather confusing. Actually, when does the difference in Sox17-Erg- and Etv2- induced ECs start to appear?

Thank you for bringing up this point. Our initial scRNAseq UMAP plot indicates that the difference arises as soon as Day 3 (Figure 4b) as seen by the differential clustering of the sample types. We believe that difference in cell cycle status at Day 3 further demonstrates the different mechanism they use to achieve their different end point identity; potentially arterial-like in the case of Sox17-Erg. At the individual gene level, we see start to see differences between Sox17-Erg and Etv2 at Stage#2 which is further amplified at Stage#3. The similarity of GO-terms for Stage#1 and Stage#2 highlights that despite the differences in individual gene expression that the starting themes of reprogramming may be similar between the two methodologies but results in different end-points. This is reinforced by our Day 3 CUT&Tag data which highlights similar transcription factor motifs. To convey this point better to the reader we modified our main text about Figure 5a-f to read: "Despite the similar GO-term enrichment, the trajectories diverged by Stage#2 seen by cluster-specific expression of module genes (Fig. 5e, 5f)."

9. It is somewhat acceptable that Sox17-Erg-generated ECs are arterial-like, and Etv2-induced ones are venous or lymphatic-like. But still the evidence is scarce. Not just several markers (i.e. Gja5, VEGFR3 etc.) but more comprehensive analysis comparing with endogenous primary ECs would be required for more convincing conclusion.

We have now provided qPCR data comparing adult murine cardiac fibroblasts to adult Etv2 iECs, adult SEG iECs, and isolated native adult murine heart endothelial cells (**Extended Data Figure 11**). Since the native heart endothelial cells are heterogeneous, we expect there to be expression of both arterial (Notch1, Dll4, Gja5, Hey1) and venous (Nr2f2 and vWF) markers in

the native heart ECs and minimal levels in the control fibroblasts. At Day 3, we see elevated levels of all arterial markers and Nr2f2 relative to the control sample in the SEG iECs and the arterial marker levels persist into Day 7. Interestingly for Dll4, Gja5, and Hey1, the expression of these genes in SEG samples are similar or elevated beyond the heterogeneous native heart endothelial cells suggesting Sox17-Erg iECs are arterial-like. We do see a slight increase in Nr2f2 levels at Day 3 and higher levels of vWF at Day 7 in SEG samples compared to control samples but those levels are below the native ECs. The sustained levels of Vegfr3, Nr2f2 and vWF found in the Etv2-samples at similar or lower levels to native endothelial cells do point to these cells being venous, lymphatic, or progenitor-like. We acknowledge that more detailed follow up experiments would need to be done to provide a more complete picture, however, this additional qPCR data does further support our conclusions regarding the Sox17-Erg iEC arterial-like identity.

10. Endogenous Etv2 is expressed in a progenitor stage of ECs but not in ECs. Continuous expression of Etv2 after the conversion to ECs would be considered to hinder EC maturation. So, it would be likely that the maturation status of Sox17-Erg-generated ECs is relatively higher than that of Etv2-induced ECs. But that would not mean that Sox17-Erg-generated ECs are mature ECs.

We agree that there are caveats for our defining SEG iECs as completely mature ECs. However, the heterogeneity of mature cardiac endothelium and our of reprogrammed iECs makes it hard to define what is an unequivocally mature endothelial cells. For example, the markers expressed by the cardiac endothelial cell progenitors during cardiac development (i.e. Nr2f2 and Vegfr3) are also markers for venous (Nr2f2) or lymphatic (Vegfr3) adult endothelium, and the relative functional properties (i.e. shear stress tolerance) varies between endothelial cell type. We define SEG iECs as mature since Sox17-Erg iECs on the whole display molecular and functional properties that are found in adult mature endothelial cells. We have added a few sentences to our Discussion section to provide further context and discuss the caveats of SEG maturation status.

Reviewer #3

1) The claim of that the iECs display a significantly better functional property of endothelial cells, a better interact with cardiomyocytes, and better engraft in mouse model of AMI, are significant ones. You need to demonstrate these findings, and describe each of them quantitatively in the Abstract.

We have addressed this comment with two sets of experiments. First, we reperformed the co-culture assay to include quantification of Gja1 gap junction interaction between the cardiomyocytes and the other cultured cell type (fibroblasts, iECs, or native heart endothelial cells). This data indicates that we do not see a statistical difference in the interactions between the cardiomyocytes and the other cell types, which we interpret as iECs preserve their ability to interact with cardiomyocytes *in vitro* after the reprogramming process (Figure 3e,f). We have updated our main text to reflect this new data. To reflect the functional implications of iEC injection

Revision Figure 10: Cardiac function and scar region perfusion analysis of four week samples. a) Fractional Shortening of samples pre- and post-infarction. b) Percent area of perfused vessels found in scar region relative to normal non-scar area.

in the AMI model, we have collected echocardiography and lectin perfusion data of four week samples. The collected echocardiography does not show improvement of cardiac function in the iEC samples when compared to the PBS controls (**Figure 3m** and **Revision Figure 10a**). This lack of difference is not surprising since we are injecting new cells into the heart and are not reprogramming the in situ fibroblasts. However, we do report a quantified increase in the area of perfused vessels in the iEC controls compared to the PBS controls at the four week timepoint (**Figure 3n-p** and **Revision Figure 10b**). We have updated our main text and abstract to reflect these *in vivo* findings.

2) Likewise, the claim that iECs are more arterial in nature (compared to ETV2 derived?) need quantitative data to support.

We have now added qPCR data comparing SEG iECs to control fibroblasts, Etv2 iECs, and isolated adult murine heart endothelial cells (**Extended Data Figure 11**). SEG iECs express higher levels of the arterial marker genes *Gja5* and *Hey1* at both Day 3 and Day 7 of reprogramming when compared to control fibroblasts, while Etv2 iECs do not. The relative levels of *Dll4*, *Gja5*, and *Hey1* in the iEC samples are all similar or greater than the native heart endothelial cell samples; which would contain a heterogeneous population of endothelial cells (including artery and venous cells).

3) The conclusion of that iECs are more mature endothelial state than the traditional Etv2 reprogramming derived ECs also need the quantitative data to support, which need to be described in the abstract. Why it is important? The pros and cons of mature vs immature ECs should be discussed in Discussion.

Thank you for this comment, we have included new data and new text in the Discussion section to discuss the different potentials of reprogrammed immature and mature endothelial cells. In the newly provided qPCR data (**Extended Data Figure 11**), we see a higher level of *Vegfr3* in Day 3 Etv2 iECs than all other sample types and a maintained higher level of *Nr2f2* when compared to control samples. SEG iEC qPCR data, on the other hand, do not appear to express either marker (*Nr2f2* or *Vegfr3*) at significant levels and transition to expressing arterial endothelial cell markers. Together, we interpret this data to mean that the SEG iECs have targeted an arterial-like identity while Etv2 iECs target a progenitor state.

Reviewer #4

The major criticism is that these studies lack new mechanistic insights and is primarily a descriptive study. The *in vivo* studies add very limited new insights—the study that should have been performed is a fate mapping approach with the delivery of Sox17-Erg into the heart following injury that demonstrates a functional improvement in cardiac function. Other minor issues are outlined below in no particular order.

Thank you for your comments and feedback. We acknowledge that our overall study focuses characterizing Sox17-iEC reprogramming and does not delve deeply into the downstream mechanism. Based on reviewer feedback, we now have provided preliminary evidence that the primary mediators of Etv2 and Sox17-Erg are different based on combined analysis of our CUT&Tag and scRNAseq data. We have also added new data that includes lineage tracing of Tcf21+ adult fibroblasts by using primary Tcf21 tdTomato+ cells *in vitro* (**Figure 4r-s** and **Revision Figure 11**) as well as functional analysis of 4 week post myocardial infarct hearts. Our echocardiography analysis of the four-week hearts revealed that the cardiac function of the

hearts did not improve when compared to PBS controls. However, lectin perfusion analysis of the scar areas revealed marked improved of the perfusion of the iEC samples than the controls. The lack of cardiac function improvement is not surprising given that the reprogrammed cells are injected into the functional site and the in situ fibroblasts are not reprogrammed. We view iEC reprogramming as a complimentary approach to iCM reprogramming and the beneficial impact on cardiac function of the injected iECs may take longer to manifest than the four-week timepoint. As your comments highlight, we view the mechanism of Sox17-Erg reprogramming and the future in vivo applications as exciting directions to explore in future studies. Please see below for more specific follow up to your comments.

1) The positive-negative selection for fibroblasts should include a negative selection for the endothelial progenitors such as Flk1 and Tie2. CD31 is an intermediate endothelial marker.

We appreciate this key point that endothelial progenitors may affect the outcome of this reprogramming. To directly address this, we added a Tcf21-CreERT2 ROSA26-tdTomato lineage tracing model to this manuscript. In brief, we labeled adult murine Tcf21-positive cells (a fibroblast-specific population of cells) with TdTomato via tamoxifen injection and then isolated cardiac fibroblasts using our adult fibroblast isolation methodology. We then performed iEC reprogramming and analyzed Day 7 cells using flow cytometry using Pecam as a marker for endothelial cells (**Figure 4r,s** and **Revision Figure 11**). Our results demonstrate a statistically significant increase in the presence of Tcf21-tdTomato+ Pecam+ cells in our SEG condition (average of 17.73%) compared to the negative control (average of 0.28%).

Revision Figure 11: Flow Cytometry of Day 7 Tcf21-tdTomato labeled iECs. a) representative flow cytometry data. b) quantification of percent Tcf21-tdTomato+ Pecam1+ cells. (**** p-value < 0.0001)

2) In figure 1 g-h, the authors should plot the actual %-Pecam-positive cells rather than normalization to the 10 factors (used to initiate the assay). This will help gauge the overall efficiency of the reprogramming. Moreover the quantification should be performed using both FACS and immunohistochemistry. For example, panel 1n does not correlate with any of the quantitative data.

We have changed Figure 1g-j to reflect the absolute percentage of Pecam-positive cells. We decided to use immunofluorescence to initially gauge reprogramming efficiency in order to take cell morphology and interaction (like cord formation on collagen matrix) into account as an indicator of endothelial cell reprogramming for our screening approach. We acknowledge that flow cytometry in addition to the immunohistochemistry would have been ideal, however, we feel

that our follow-up characterization and transcriptomic experiments corroborate our findings in the initial factor screening.

3) As outlined in Figure 2, the authors should provide a time course analysis of the reprogramming studies that were performed up to 4 weeks. Quantification of the endothelial reprogrammed cells over the specified time course will help determine whether the endothelial cells continue to increase or reach a plateau after a certain time period during the reprogramming process.

We have added Cdh5 Pecam dual labeled flow cytometry of one week, two week, and four week old samples to analyze how the composition of the reprogrammed cells change over time (Figure 2b-e and Revision Figure 12 for an abridged version of data found in main manuscript). We do see a decrease in the number of double positive cells over time in our Sox17-Erg sample, similar to what is found in the literature with Etv2-based reprogramming approaches.

Revision Figure 12: Flow Cytometry Cdh5 Pecam Dual Staining. a) Representative Day 7 data of control, Etv2, and Sox17-Erg conditions. b) Graph of % Pecam+ Cdh5+ of Day 7, 2 weeks, and 4 weeks samples (** p-value < 0.01, *** p-value < 0.001)

4) The GO Biological Processes provided in Figure 2 should be placed in the supplemental data section.

As per your comment, the GO Biological Process data has been moved to Extended Data Figure 4a.

5) Although the study has undertaken several functional assays for the reprogrammed endothelial cells, the authors should include tube formation assays.

We have added tube formation assay as one of our functional assays and our SEG iECs show comparable tube formation to that of native Heart Endothelial cells after 4hrs of culture on Matrigel (Extended Data Figure 4d and Revision Figure 13).

Revision Figure 13: Cord Formation Assay (4hrs after plating)

6) The authors report a difference in the efficiency of the

endothelial reprogramming while using the polycistronic constructs. Did they look for an actual difference in the expression of Sox17 and ERG depending on their position in the cassette?

We performed and now included qPCR analysis of the two different polycistronic constructs (**Extended Data Figure 4b** and **Revision Figure 14**). We do see a statistically significant difference in expression of both Sox17 and Erg depending on the position. According to our qPCR results, we see that ESG results in higher expression of both Sox17 and Erg. However, we do see a larger increase in Erg expression in the ESG construct when compared to the increase in Sox17 expression.

Revision Figure 14: qPCR analysis of Sox17-Erg polycistronic constructs

7) As outlined above, the *in vivo* studies are really a major detraction for this manuscript. While the authors have performed myocardial infarction injuries and have injected the iECs to demonstrate engraftment, did they perform any assays to assess whether this engraftment leads to an improvement of the cardiac function in the hearts injected with iECs vs. control infarcted animals.

We have now included four week echocardiography and lectin perfusion data that shows no change in overall cardiac function but an improvement in the perfusion of the vessels found in the scar area (**Figure 3m-p** and **Revision Figure 15**). The lack of cardiac functional improvement when compared to PBS controls is not surprising since we did not target the *in situ* fibroblasts, and the impact of iECs on the cardiac functional may take longer to manifest than iCMs.

Revision Figure 15: Cardiac function and scar region perfusion analysis of four week samples. a) Fractional Shortening of samples pre- and post-infarction. b) Percent area of perfused vessels found in scar region relative to normal non-scar area.

8) Did the authors evaluate the potential of only the Pecam-positive population compared to the entire population of fibroblasts undergoing reprogramming?

This is a very interesting question that we hope to explore in future studies. We decided to use the heterogeneous reprogramming population to see if the Pecam-positive cells are present in the Day 7 samples, and now have added a four-week timepoint to see if the injection of the Sox17-Erg iECs result in functional changes.

9) Regarding the single cell analyses performed, were there any insights regarding the molecular programs that were uniquely governed by Sox17-Erg vs. ETV2? Does Sox17-Erg induce ETV2 expression or Flk1 expression? Is Sox17-Erg a pioneer factor complex?

Our scRNAseq data provides initial insights that the differences in the molecular programs of these two reprogramming methodologies is apparent at Day 3. To further test whether ETV2 or Sox17-Erg induce the expression of factors of the opposing cocktail, we have added qPCR data that compares the transcription levels of these genes at Day 3 and Day 7 (**Extended Data Figure 11a and 11b**). Interestingly, the transcription level ETV2 is not altered in the Sox17-Erg sample when compared to control fibroblasts. We do see an increase in the levels of Erg in the ETV2 iEC samples at both Day 3 and Day 7, but not Sox17. Of note, the transcription levels of

Erg in the Etv2 levels does not reach that of native adult murine endothelial cells. Both Etv2 and Sox17-Erg induces the increased expression of Flk1 at both Day 3 and Day 7 (**Extended Data Figure 11a and 11b**). Together, these results suggest that perhaps Erg plays a role, at some level, in both reprogramming methodologies. The new qPCR data does show that Etv2 also induces the higher expression of Fli1 when compared to fibroblasts while Sox17-Erg does not. This is of note due to recent work highlighting the compensatory mechanisms of Erg and Fli1 in endothelial homeostasis³.

We do plan on further evaluating potential for Sox17-Erg being a pioneer complex in future manuscripts that further delve into the downstream mechanism of Sox17-Erg.

10) To distinguish the mechanistic differences between the different networks governed by Etv2 vs Sox17-Erg, ChIP-Seq or Cut & Run in combination with ATAC-Seq analysis and the scRNASeq dataset will allow the authors to identify the different gene sets activated by the respective method of reprogramming in the cardiac fibroblasts.

To gain further insight into the differential mechanism of Etv2 and Sox17-Erg, we first provided the new qPCR data mentioned above to show that there is potential overlap based on induced Erg expression in the Etv2 samples, but the increased transcription levels of Erg does not match neither SEG nor native adult murine endothelial cells (**Extended Data Figure 11a and 11b**). Additionally, Etv2 reprogramming results in the increase of the transcription of Fli1 relative to control samples while SEG iECs have lower levels of Fli1 relative to control samples. Since Fli1 was one of the candidate transcription factors for the Etv2 H3K27ac differentially accessible peaks, we have included a plot that illustrates the scRNAseq expression of a few top candidate H3K27ac transcription factors for both SEG and Etv2 (**Extended Data Figure 11c**). As seen with Fli1 in the qPCR data, Fli1 is more highly expressed in the Etv2 cells than SEG iECs. This trend is also seen for the transcription factor Rela (a downstream transcription factor candidate for Etv2 H3K27ac peaks). Sox17-Erg iECs appear to increase in the expression of Jun, Junb, and Ets1 (candidates for SEG H3K27ac peaks) from Day 3 to Day 7. Etv2 samples show similar trends for Jun and Junb, but lower expression of Ets1. Overall, this data provides preliminary evidence of a reliance and complimentary increase in expression of different downstream transcription factors for acquisition of the two targeted cell fates and, perhaps, different mechanisms for the maintenance of the acquired iEC identity³. We have added a sentence in our H3K27ac results section describing this difference. We acknowledge this is only a cursory exploration into this interesting question and plan on further exploring this downstream mediators of Sox17-Erg direct reprogramming in future manuscripts.

Bibliography

1. Zhou, Y. *et al.* Single-Cell Transcriptomic Analyses of Cell Fate Transitions during Human Cardiac Reprogramming. *Cell Stem Cell* **25**, 149-164.e9 (2019).
2. Trimm, E. & Red-Horse, K. Vascular endothelial cell development and diversity. *Nat. Rev. Cardiol.* 1 doi:10.1038/s41569-022-00770-1.

3. Gomez-Salinerro, J. M. *et al.* Cooperative ETS Transcription Factors Enforce Adult Endothelial Cell Fate and Cardiovascular Homeostasis. *Nat. Cardiovasc. Res.* **1**, 882–899 (2022).

REVIEWER COMMENTS

Reviewer #1 (Remarks to the Author):

The revised manuscript effectively addressed my initial critiques. The authors present a compelling manuscript describing a new approach to reprogram fibroblasts to induced endothelial cells. The only concern that I raise with this version is that the statistical analysis for comparisons of more than 2 groups uses T-tests for multiple datasets including figures 2e, 2h, 2l, 3b, 3c, 3f, 4p, and 4s. This is not the appropriate statistical analysis and should be corrected.

Reviewer #2 (Remarks to the Author):

In this revised manuscript, Farber et al. showed efforts to address the reviewer's comments. The authors have addressed the concerns in some degree, but several important core issues have not been addressed.

1. As the authors showed in Fig. 2a and Extended Figure 2, endothelial depletion step did not change the percentage of endothelial cells after the step. How do the author consider the meaning of the process?

2. Though it is unclear the details of fibroblast marking with Tcf21-Cre/loxP system as the method was not described in the Method section, FACS plots in Fig. 4r is difficult to interpret. What is the Tcf21-TdTomato-negative population? Comparing the middle panel (Etv2) and right panel (Sox17-Erg-GFP), all cell populations are uniformly moving upward in the right panel. All the three subpopulations in the TdTomato-negative cells in the middle panel uniformly shift upward. TdTomato-positive population in the middle panel is also moving upward in the right panel with never forming Pecam1-positive distinct endothelial cell population. The reviewer would like to ask to explain the situation scientifically.

3. In Fig. 2d, Sox17-Erg treatment induced Pecam1-single positive and Cdh5-single positive cells. Pecam1 and Cdh5 expressions are mutually exclusive. What are these cell populations? The authors mentioned that arterial EC-like cells would be induced in this study. But arterial ECs reported previously are all Pecam1/Cdh5 double positive.

4. Fig. 3n-p: the perfusion of the scar area should be also shown in lower magnification views to describe the gross appearance of the perfusion of the heart. Higher magnification views alone like Fig. 3o and 3p

may not reflect the truth. In addition, the perfusion percentages of Fig. 3n are shown as relative to “nonMI” area. It is somewhat unreasonable to use for the main data showing the effects. Honestly, the authors showed the not relative percentages of perfusion area in Extended Figure 5b. This graph apparently shows that the perfusion percentage area in MI region was not different between iEC injection and control (PBS injection). Perfusion in non-MI region are somewhat different. So, by showing the results with the relative value to “nonMI” area, it seems to become possible to show some significance as shown in Fig. 3n. Such manipulation does not tell the truth.

It may be acceptable that this method is one of the direct reprogramming methods that can induce endothelial cells. Nevertheless, many advanced points that the authors raised, high efficiency of endothelial cell induction, induction of “cardiac” endothelial cells, and efficient cardiac repair (than other endothelial cells), have not been sufficiently supported by data shown in this study.

Reviewer #3 (Remarks to the Author):

responses are satisfactory. manuscript has been significantly improved.

Reviewer #4 (Remarks to the Author):

The authors are only partially responsive regarding the issues raised in the initial review. For publication in this journal, a clear mechanism should be defined, in vivo fate mapping should be performed to demonstrate in vivo reprogramming and scRNA-seq should be performed at all time periods due to the heterogeneity of the reprogramming process.

Thank you for the insights and comments from the reviewers and editors involved with this manuscript. For this revision, we have included two new figures (**Figure 6** and **Figure 7**) and updated the previous version of the manuscript in response to reviewer comments. **Figure 6** is scRNAseq analysis of adult murine cardiac fibroblast iEC reprogramming where we examine the reprogramming heterogeneity and, in combination with our previous H3K27ac data, preliminarily identify *Elk3* as a downstream transcription factor mediator of the early stages of the Sox17-Erg reprogramming process. **Figure 7** is Sox17-Erg *in vivo* direct reprogramming using Tcf21-iEC tdTomato mice where we demonstrate the direct conversion of Tcf21-positive Pecam1-negative cells into Pecam1-positive cells.

Please see below for more details on how we addressed specific comments.

Reviewer #1 (Remarks to the Author):

The revised manuscript effectively addressed my initial critiques. The authors present a compelling manuscript describing a new approach to reprogram fibroblasts to induced endothelial cells. The only concern that I raise with this version is that the statistical analysis for comparisons of more than 2 groups uses T-tests for multiple datasets including figures 2e, 2h, 2l, 3b, 3c, 3f, 4p, and 4s. This is not the appropriate statistical analysis and should be corrected.

We have amended our previously incorrect statistical analyses to the appropriate one way ANOVA with Tukey post test.

Reviewer #2 (Remarks to the Author):

In this revised manuscript, Farber et al. showed efforts to address the reviewer's comments. The authors have addressed the concerns in some degree, but several important core issues have not been addressed.

1. As the authors showed in Fig. 2a and Extended Figure 2, endothelial depletion step did not change the percentage of endothelial cells after the step. How do the author consider the meaning of the process?

While the depletion step did not reduce the level of Pecam1-positive cells to statistical significance, we have continued to include this step in our isolation of fibroblasts to ensure consistency between different experiments and to ensure that our isolated starting cell population has as few Pecam1-expressing cells as possible.

2. Though it is unclear the details of fibroblast marking with Tcf21-Cre/loxP system as the method was not described in the Method section, FACS plots in Fig. 4r is difficult to interpret. What is the Tcf21-tdTomato-negative population? Comparing the middle panel (Etv2) and right panel (Sox17-Erg-GFP), all cell populations are uniformly moving upward in the right panel. All the three subpopulations in the tdTomato-negative cells in the middle panel uniformly shift upward. TdTomato-positive population in the middle

panel is also moving upward in the right panel with never forming Pecam1-positive distinct endothelial cell population. The reviewer would like to ask to explain the situation scientifically.

We apologize for neglecting to include our methodology of Tcf21-CreERT2 tdTomato (Tcf21-iCre) tamoxifen induction, which would have provided better clarity for this experiment. We have now included this methodology in our methods section. In short, we induce Tcf21-iCre expression via five daily doses of tamoxifen in adult mice (6-12 weeks of age) resulting in the labeling of Tcf21-positive cells. To date, the Tcf21 tdTomato model is the best available methodology to label *in vivo* a large population of adult cardiac fibroblasts and has been used in other direct reprogramming studies. This Tcf21-iCre has been shown to primarily label fibroblasts and does not label endothelial cells¹. However, Tcf21-iCre is known to not label all adult cardiac fibroblasts and Thy1 expression does not exactly mirror Tcf21 expression resulting in Thy1-positive Tcf21-negative fibroblasts in our isolated cells. Therefore, the tdTomato-negative population are the Thy1-positive Tcf21-negative cells that were isolated and then underwent iEC reprogramming. The lack of a distinct Pecam1-positive population is likely due to reprogramming heterogeneity which we see in **Figure 2b**, **Figure 6**, and **Extended Data Fig. 12c,d** (please see comment#3 for further discussion).

3. In Fig. 2d, Sox17-Erg treatment induced Pecam1-single positive and Cdh5-single positive cells. Pecam1 and Cdh5 expressions are mutually exclusive. What are these cell populations? The authors mentioned that arterial EC-like cells would be induced in this study. But arterial ECs reported previously are all Pecam1/Cdh5 double positive.

In this revised manuscript, we have included our newly generated scRNA-seq data of adult iECs (**Figure 6**) which shows that the heterogeneity of the Sox17-Erg iECs changes over time. This includes the co-expression of Cdh5 and Pecam1 (**Extended Data Figure 12d** and **Revision Figure 1**). We agree that arterial endothelial cells are positive for both markers, and both the flow cytometry analysis and adult iEC scRNAseq analysis show a decrease in the endothelial-like cells at the four week timepoint when compared to the earlier timepoints (**Figure 6c**). These single marker positive cells are likely partially reprogrammed iECs (pre-iECs), which are similar to pre-iCMs that we and others have observed during iCM reprogramming^{2,3}. We acknowledge that for most occasions pre-iECs are not optimal for translational or therapeutic use, and would recommend sorting for double marker positive cells when needed.

Revision Figure 1: scRNAseq Gene Expression Scatter Plot of Cdh5 and Pecam1.

4. Fig. 3n-p: the perfusion of the scar area should be also shown in lower magnification views to describe the gross appearance of the perfusion of the heart. Higher magnification views alone like Fig. 3o and 3p may not reflect the truth. In addition, the perfusion percentages of Fig. 3n are shown as relative to “nonMI” area. It is somewhat unreasonable to use for the main data showing the effects. Honestly, the authors showed the not relative percentages of perfusion area in Extended Figure 5b. This graph apparently shows that the perfusion percentage area in MI region was not different between iEC injection and control (PBS injection). Perfusion in non-MI region are somewhat different. So, by showing the results with the relative value to “nonMI” area, it seems to become possible to show some significance as shown in Fig. 3n. Such manipulation does not tell the truth.

We apologize for the confusion. To clarify, we have expanded and revised our methods section to provide greater detail on how we imaged and quantified our samples. In the previous version of our manuscript, we provided the relative perfusion area (MI versus nonMI) because the induced scar can impact the overall perfusion of the heart and we wanted to take that into account since LAD ligation surgery can lead to variable scar sizes despite our best efforts. Meanwhile, we also feel this reviewer’s concern is valid. Therefore, based on this feedback and with our intention to be as transparent as possible to all readers, we now also report the scar perfusion area and provide the low magnification stitched scans of the infarct areas (**Figure 3n, o**). In our revised methodology, we image the whole scar area per section and perform blinded quantification of the scar perfusion area. This new analysis led to a trend of increased perfusion in the iEC samples (average area of 4.99% for PBS and 10.83% for iEC), and we have updated the text of this manuscript to reflect this new analysis (**Revision Figure 2**).

Revision Figure 2: Quantification of 4 week sample lectin perfusion of MI scar area

It may be acceptable that this method is one of the direct reprogramming methods that can induce endothelial cells. Nevertheless, many advanced points that the authors raised, high efficiency of endothelial cell induction, induction of “cardiac” endothelial cells, and efficient cardiac repair (than other endothelial cells), have not been sufficiently supported by data shown in this study.

We appreciate that the reviewer acknowledged the value of our new methodology to reprogramming fibroblasts into iECs. We also agree that many interesting directions can be followed upon in the future, including the ones suggested by this reviewer. We have now included such discussion in our revised manuscript.

1. Acharya, A., Baek, S. T., Banfi, S., Eskiocak, B. & Tallquist, M. D. Efficient Inducible Cre-Mediated Recombination in Tcf21 Cell Lineages in the Heart and Kidney. *Genesis* **49**, 870–877 (2011).
2. Liu, Z. *et al.* Single-cell transcriptomics reconstructs fate conversion from fibroblast to cardiomyocyte. *Nature* **551**, 100–104 (2017).
3. Ieda, M. *et al.* Direct Reprogramming of Fibroblasts into Functional Cardiomyocytes by Defined Factors. *Cell* **142**, 375–386 (2010).

Reviewer #3 (Remarks to the Author):

Responses are satisfactory. Manuscript has been significantly improved.

We are pleased that this reviewer is satisfied with our revised manuscript.

Reviewer #4 (Remarks to the Author):

The authors are only partially responsive regarding the issues raised in the initial review. For publication in this journal, a clear mechanism should be defined, *in vivo* fate mapping should be performed to demonstrate *in vivo* reprogramming and scRNA-seq should be performed at all time periods due to the heterogeneity of the reprogramming process.

In this revised manuscript, we added our newly generated scRNAseq of Day 3, Day 7, 2 weeks, and 4 weeks of age iECs derived from adult murine cardiac fibroblast (**Figure 6**) and *in vivo* direct reprogramming using the most stringent genetic lineage tracing approach (**Figure 7**). Over the last three months, we have spent tremendous efforts on optimizing the procedure, collecting various sets of tissue or cell samples, and analyzing the data for these two figures as each experiment had a unique set of challenges. We believe these two sets of new data have not only addressed the concerns from the reviewer but also significantly strengthened the manuscript in its novelty, potential impact and thoroughness. We appreciate the suggestions from this reviewer.

Fig. 6

Fig. 6- aiEC heterogeneity changes during reprogramming **a**, UMAP of aiEC scRNAseq labeled by sample type. **b**, UMAP of aiEC scRNAseq labeled by cluster name. **c**, Percent cluster composition of samples. Top three aiEC-related cluster percentages are labeled for SEG sample. **d**, Expression profiles of fibroblast and endothelial genes. **e**, Top ten marker gene heatmap for each cluster. **f**, Cluster-specific expression of reprogramming and endothelial genes. **g**, Heatmap of top ten identified gene regulatory network regulons for selected aiEC endothelial-like clusters. **h**, Expression profile of *Eik3*. **i**, Venn diagram of aiEC conserved marker genes genes and *Eik3* regulon genes. **j**, Venn diagram of *Eik3* regulon genes, H3K27ac SEG upregulated genes in neonatal iECs, and H3K27ac neonatal heart endothelial cell genes. **k**, GO Biological Process Enrichment analysis of *Eik3* regulon to H3K27ac HEC and SEG, and aiEC conserved genes.

The new scRNAseq data highlights the changes in iEC heterogeneity with four clusters being identified as endothelial-like and sample composition percentage of those clusters changing over time (**Fig. 6c**). We performed gene regulatory network analysis on these

four clusters to preliminarily identify the downstream transcription factors that govern the reprogramming heterogeneity and found Elk3 as early mediator of Sox17-Erg iEC reprogramming process. Further analysis of the identified Elk3 regulon genes to our Day 3 H3K27ac data highlights the role of Elk3 and its regulated genes in the acquisition of endothelial identity in Sox17-Erg iEC reprogramming (**Fig. 6g-k**, and **Revision Figure 3**).

Revision Figure 3: Elk3 regulon linked to early stages of Sox17-Erg iEC reprogramming. **a**, Gene expression plot of Elk3. **b**, Venn diagram of Elk3 regulon, and Day 3 H3K27ac upregulated genes from neonatal SEG reprogramming and neonatal heart endothelial cells (HEC). **c**, GO Biological Process Enrichment Analysis of Elk3 regulon, Day 3 H3K27ac upregulated genes, and conserved adult fibroblast iEC marker genes.

Fig. 7

Figure 7 contains the newly added Day 7 *in vivo* iEC reprogramming where we demonstrate the conversion of Tcf21-tdTomato fibroblasts into Pecam1 expressing cells in infarcted hearts injected with Sox17-Erg-GFP retrovirus (**Figure 7**).

Fig. 7- a, Schematic of *in vivo* iEC reprogramming. **b**, Quantification of percent Pecam⁺ GFP⁺ Tcf21-tdTomato⁺ of GFP⁺ Tcf21-tdTomato⁺ cells in Day 7 hearts (Mann-Whitney test). Two representative low magnification images of Day 7 GFP control (**c**) and SEG (**d**) samples (scale bars 100 μ m). Higher magnification images of GFP (**e**) and SEG (**f**) samples (scale bars 20 μ m). Yellow arrows indicate GFP⁺ tdTomato⁺ Pecam⁻ cells. Blue arrows indicate GFP⁺ tdTomato⁺ Pecam1⁺ cells.

REVIEWER COMMENTS

Reviewer #1 (Remarks to the Author):

The authors have addressed all of my concerns.

Reviewer #2 (Remarks to the Author):

In the second revised manuscript by Farber et al., it appears that some of the key concerns previously raised have not been fully addressed.

1. In the revised manuscript, the reviewer noticed that the concerns raised in the previous comment #2 have not been fully addressed. Specifically, the reviewer had asked for a scientific explanation regarding the observations in Figure 4r, the three peaks of subpopulations in the tdTomato-negative cells (around 10^3 , $1-2 \times 10^2$, and less than 0) and the single-peaked tdTomato-positive population in the middle panel (Etv2) seemed to uniformly move upward in the right panel (Sox17-Erg-GFP) (around 5×10^3 , 10^3 , and 3×10^2). Additionally, the appearance of a distinct single-peaked PECAM1-positive endothelial cell population, as suggested in immunostaining (Figure 4q), was not evident in the Sox17-Erg-GFP panel of Figure 4r. This discrepancy might indicate non-specific staining (or maybe not quite suitable signal amplitude setting) in FACS analysis, rather than the specific emergence of PECAM1-positive endothelial cells. Could you please provide further clarification or additional data to address this point?

2. Regarding the authors' response to the previous comment #3, the reviewer observed that in Figure 6, the expressions of Cdh5 and Pecam1 largely overlap in SEG at 4 weeks (Figure 6a and 6d). This observation seems to be in contrast with the results shown in Figure 2d. Could you please elaborate on this apparent discrepancy?

3. In response to the previous comment #4, the reviewer appreciates the inclusion of lower magnification views in Figure 3n and o. However, these still appear to be selective views. Typically, whole appearances of infarct hearts with multiple sections are presented. Moreover, it remains unclear how iECs contribute to the formation of lectin/GFP-double positive, perfused, functional vessels. Could you provide additional information or evidence to clarify this aspect?

4. In the Abstract, the authors argue that Sox17-Erg reprogramming directs cardiac fibroblasts towards a differentiated arterial-like identity, leading to a more efficient and mature endothelial cell conversion compared to traditional Etv2-based reprogramming. However, it appears that this conclusion is primarily based on a very limited set of gene expression data, being insufficient to robustly support your main arguments.

Overall, the authors' arguments seemed to not fully supported with solid and convincing data in this study.

Reviewer #4 (Remarks to the Author):

The authors have performed additional studies but in its current form the manuscript lacks a clear mechanism and its data are too preliminary. The authors have established expertise and have contributed extensively to the field of cardiomyocyte reprogramming so it is unclear why the studies in Figure 7 are so preliminary. As the authors are well aware, with any myocardial injury study the following should be demonstrated:

- 1) use of serial echocardiography to determine whether Sox 17 reprogramming improves cardiac function over time (D7, D30, D60). These are standard studies and analyses for the field.
- 2) More convincing tdTomato-GFP coexpression in vasculature should be shown and not just in individual cells embedded within the ECM. Perhaps these results are due to the analyses being performed so early following injury/delivery of the retrovirus.
- 3) bulk RNA-seq to compare the results of in vivo reprogramming vs. control virus
- 4) With regards to Figure 6, the single cell analyses helped define a Sox17-Elk3 cascade but how important is Elk3? Elk3 is regulated by many transcription factors including Etv2. Perhaps the study should be performed in an Elk3 deficient line using gene editing or knockdown strategies. The single cell studies should have also been compared to Etv2 mediated reprogramming. The mechanism whereby Etv2 functions has been defined and an advance for the field would be to understand how Sox17 mediates the reprogramming? Does it function via super enhancers or is it a pioneer factor or does it squelch repressors, etc.? Further, the description and explanation of the data in Figure 6 is too superficial. Just to list the top ten genes in cell clusters (Fig. 6e) is not sufficient to address a mechanism of action.

Reviewer #2 (Remarks to the Author):

1. In the revised manuscript, the reviewer noticed that the concerns raised in the previous comment #2 have not been fully addressed. Specifically, the reviewer had asked for a scientific explanation regarding the observations in Figure 4r, the three peaks of subpopulations in the tdTomato-negative cells (around 10^3 , $1-2 \times 10^2$, and less than 0) and the single-peaked tdTomato-positive population in the middle panel (Etv2) seemed to uniformly move upward in the right panel (Sox17-Erg-GFP) (around 5×10^3 , 10^3 , and 3×10^2). Additionally, the appearance of a distinct single-peaked PECAM1-positive endothelial cell population, as suggested in immunostaining (Figure 4q), was not evident in the Sox17-Erg-GFP panel of Figure 4r. This discrepancy might indicate non-specific staining (or maybe not quite suitable signal amplitude setting) in FACS analysis, rather than the specific emergence of PECAM1-positive endothelial cells. Could you please provide further clarification or additional data to address this point?

We apologize for missing the opportunity to further explain the technical details. For our flow experiment, the gating strategy is based on the knowledge found in the literature that the control condition has few if any Pecam+ TdTomato+ cells since Tcf21-tdTomato labels a Pecam-negative (non-endothelial) population of Tcf21-positive cells. This is further confirmed in our in vivo experiments found in Figure 7. Therefore, the potential peaks below the control sample gating are viewed as Pecam-negative populations. Figure 4p and q show immunofluorescence quantification and representative image based on a minimum threshold and are not as nuanced flow cytometry in representing cell-specific protein level. However, the presented images do show iECs with different intensities of PECAM1 expression. Our flow cytometry and immunofluorescence data match in showing a higher number of PECAM1+ cells in SEG samples when compared to both Etv2 and control samples.

2. Regarding the authors' response to the previous comment #3, the reviewer observed that in Figure 6, the expressions of Cdh5 and Pecam1 largely overlap in SEG at 4 weeks (Figure 6a and 6d). This observation seems to be in contrast with the results shown in Figure 2d. Could you please elaborate on this apparent discrepancy?

The difference is due to the different experimental conditions: the experiments in Figure 2 are using two single gene constructs (Sox17 and Erg) that are combined to infect neonatal cardiac fibroblasts at the same timepoint. The experiments in Figure 6 utilize adult cardiac fibroblasts and the Sox17-Erg-GFP polycistronic construct. Using different starting cell types as well as single versus polycistronic construct would affect the resulting efficiency and number of generated cell type, as well documented in other reprogramming fields. Additionally, the number of Pecam1+ Cdh5+ cells are not expected to be identical in these two different experimental modalities due to flow cytometry reflecting protein expression and scRNAseq reflecting transcriptome. Such differences may reflect interesting biology that worth follow-up in the future.

3. In response to the previous comment #4, the reviewer appreciates the inclusion of lower magnification views in Figure 3n and o. However, these still appear to be selective views. Typically, whole appearances of infarct hearts with multiple sections are presented. Moreover, it remains unclear how iECs contribute to the formation of lectin/GFP-double positive, perfused, functional vessels. Could you provide additional information or evidence to clarify this aspect?

We appreciate the carefulness and thoughtfulness from this reviewer. We agree that additional details and images will help strengthen the presentation. Now as we describe in the methods section, each mouse is represented by three sections, and we labeled the mouse sample that

each representative image comes from in Figure 3n so that the audience understands the context of each image. We now also provide additional images from different animals in Supplementary Figure 5e.

The GFP/lectin experiment is an interesting one, however, our focus for this experiment is lectin perfusion in the scar area and overall cardiac function to illustrate the general effects of the iEC injection. We agree that it is important to understand how iECs contribute to the formation of perfused, functional vessels and studying the interaction between iECs with the surrounding cardiac scar microenvironment will be an interesting direction in future studies and have now included in the Discussion: *“However, further work needs to be done to determine how these cells directly affect the perfusion of the scar regions.”*

4. In the Abstract, the authors argue that Sox17-Erg reprogramming directs cardiac fibroblasts towards a differentiated arterial-like identity, leading to a more efficient and mature endothelial cell conversion compared to traditional Etv2-based reprogramming. However, it appears that this conclusion is primarily based on a very limited set of gene expression data, being insufficient to robustly support your main arguments.

We have toned down our abstract to read *“Furthermore, we use genomic analyses to illustrate that Sox17-Erg reprogramming instructs cardiac fibroblasts toward an arterial-like identity. This results in a more efficient direct conversion of fibroblasts into EC-like cells when compared to traditional Etv2-based reprogramming.”* We have deleted *“as well as a more mature endothelial identify”* and changed *“directly instructs cardiac fibroblasts to a differentiated arterial-like identity”* to *“instructs cardiac fibroblasts toward an arterial-like identity”*

We have also changed our title from “Direct highly efficient endothelial reprogramming using Sox17-Erg” to “Direct conversion of cardiac fibroblasts into endothelial-like cells using Sox17 and Erg”

Reviewer #4 (Remarks to the Author):

The authors have performed additional studies but in its current form the manuscript lacks a clear mechanism and its data are too preliminary. The authors have established expertise and have contributed extensively to the field of cardiomyocyte reprogramming so it is unclear why the studies in Figure 7 are so preliminary. As the authors are well aware, with any myocardial injury study the following should be demonstrated:

- 1) use of serial echocardiography to determine whether Sox 17 reprogramming improves cardiac function over time (D7, D30, D60). These are standard studies and analyses for the field.
- 2) More convincing tdTomato-GFP coexpression in vasculature should be shown and not just in individual cells embedded within the ECM. Perhaps these results are due to the analyses being performed so early following injury/delivery of the retrovirus.
- 3) bulk RNA-seq to compare the results of in vivo reprogramming vs. control virus

We appreciate this reviewer’s excitement regarding the newly provided in vivo data. We have now included higher magnification images of the in vivo iECs (Figure 7g-h, and Supplementary Figure 13). To our knowledge, the use of the Tcf21-tdTomato lineage tracing reporter is new to the iEC field and provides proof that a non-endothelial cell is converted into a PECAM1

expressing cell which we show in our manuscript is an indicator of the acquisition of iEC identity. While we agree that serial echocardiography and transcriptomic analysis of the *in vivo* reprogramming would be important next steps for Sox17-Erg reprogramming, our *in vivo* results are just a proof of concept that Sox17-Erg can reprogram *in vivo* and the additional proposed experiments will be great follow-ups in future studies. However, to acknowledge the limitations of our study we have added this reviewer's proposed experiments as logical follow up studies in our Discussion section:

"This study also highlights the potential for this reprogramming to be used in situ, however, more in depth work needs to be done studying the identity of these in vivo reprogrammed cells and the impact of this reprogramming on non-endothelial cardiac repair since we did not evaluate heart function after in vivo reprogramming. Tracking the impact of in vivo iECs on cardiac function post-injury and analyzing how similar the in vivo reprogramming process is to the in vitro one through a combination of histological, cellular and multi-omics analyses would further develop this new reprogramming approach."

4) With regards to Figure 6, the single cell analyses helped define a Sox17-Elk3 cascade but how important is Elk3? Elk3 is regulated by many transcription factors including Etv2. Perhaps the study should be performed in an Elk3 deficient line using gene editing or knockdown strategies. The single cell studies should have also been compared to Etv2 mediated reprogramming. The mechanism whereby Etv2 functions has been defined and an advance for the field would be to understand how Sox17 mediates the reprogramming? Does it function via super enhancers or is it a pioneer factor or does it squelch repressors, etc.? Further, the description and explanation of the data in Figure 6 is too superficial. Just to list the top ten genes in cell clusters (Fig. 6e) is not sufficient to address a mechanism of action.

These are great suggestions and interesting ideas to further delineate the underlying mechanisms of how Sox17-Elk3 axis guides the EC fate in fibroblasts, in particular in comparison to Etv2 mediated pathways. We have discussed this topic more in this revised manuscript. To strengthen the current manuscript within the given time window for revision, we have now provided further *in silico* analyses with Elk3 functional perturbation. In the new Extended Data Figure 12e, simulated Elk3 knockout results in the identity shift of not fully committed iECs (cells with lower Cdh5 and Pecam1 expression) away from the EC-like cells. We now also show that while Etv2 also leads to increased expression of Elk3 in neonatal fibroblasts, only some of the other components of the defined SEG Elk3 regulon are activated in Etv2 iECs as seen in our scRNAseq and H3K27ac data (Extended Data 12f-i). This matches our scRNAseq and H3K27ac data of neonatal iEC reprogramming and further highlights the similarities and differences of these two reprogramming approaches. The goal for the adult cardiac fibroblast scRNAseq analysis in Figure 6 is to demonstrate the changes in iEC heterogeneity during different timepoints as well as evaluate adult cardiac fibroblast reprogramming since our goal is to have SEG reprogramming work in adult cells, and the starting cell type has been shown to impact the outcome of reprogramming.

Overall, one of the primary goals of this work beyond proving that Sox17-Erg is a viable reprogramming strategy is to provide a foundation of how Sox17-Erg reprogramming works to induce iEC identity in cardiac fibroblasts relative to Etv2. Many of the published Etv2 iEC and mechanistic studies use non-cardiac fibroblasts (such as mouse embryonic fibroblasts) and this study is the first, to our knowledge, to illustrate a detailed trajectory of both Etv2 and Sox17-Erg in cardiac iEC reprogramming and provides a powerful roadmap for future studies. We acknowledge that more detailed mechanistic insights are important next steps for Sox17-

Erg reprogramming and have now signified that as a future direction in this manuscript: “*Future studies can address what organ-specific properties are transferred during the iEC conversion, how Sox17 and Erg work synergistically to induce iEC reprogramming as well as address fundamental questions on how organ-specific vessels acquire and maintain their highly specific identities.*” Nevertheless, we believe both SEG and Etv2 approaches offer unique opportunities to reprogram ECs with its own pros and cons under specific context. We hope this work will trigger many interesting follow up studies and inspire new ideas in the newly emerged iEC field. We appreciate this reviewer’s thorough review and valuable comments and suggestions.

REVIEWERS' COMMENTS

Reviewer #2 (Remarks to the Author):

Despite the revisions made in this round, it is regrettable to note that the explanations and data presented still lack compelling logical persuasion. Specifically, the following issues remain inadequately addressed:

- 1 . In Figure 4r, both the middle panel (Etv2) and the right panel (SEG) show all populations shifting parallelly along the Y-axis, which appears unnatural. Unfortunately, there was no explanation provided for this observation.
- 2 . The significant differences between the FACS plots in Figure 2 and Figure 6 were not logically explained by the authors' rationale (difference of used constructs, two single gene constructs (Sox17 and Erg) and the Sox17-Erg-GFP polycistronic construct). This explanation does not account for the substantial discrepancies observed in the results.
- 3 . The relationship between the perfusion area (Lectin-positive region) and iECs is crucial for understanding the mechanism behind myocardial infarction improvement through iEC induction. However, no data were provided to elucidate this relationship.

r**Day 7 Tcf21-tdTomato iECs****Control****Etv2****Sox17-Erg-GFP**